# Genetic associations of protein-coding variants in venous thromboembolism

Xiao-Yu He[1,5], Bang-Sheng Wu[1,5], Liu Yang[1,5], Yu Guo[1,5], Yue-Ting Deng [1,5], Ze-Yu Li[2], Chen-Jie Fei[1], Wei-Shi Liu[1], Yi-Jun Ge [1], Jujiao Kang [2], Jianfeng Feng [2,3,4], Wei Cheng [1,2,3,4] ✉, Qiang Dong [1] ✉ & Jin-Tai Yu [1] ✉

Previous genetic studies of venous thromboembolism (VTE) have been largely limited to common variants, leaving the genetic determinants relatively incomplete. We performed an exome-wide association study of VTE among 14,723 cases and 334,315 controls. Fourteen known and four novel genes (*SRSF6*, *PHPT1*, *CGN*, and *MAP3K2*) were identified through protein-coding variants, with broad replication in the FinnGen cohort. Most genes we discovered exhibited the potential to predict future VTE events in longitudinal analysis. Notably, we provide evidence for the additive contribution of rare coding variants to known genome-wide polygenic risk in shaping VTE risk. The identified genes were enriched in pathways affecting coagulation and platelet activation, along with liver-specific expression. The pleiotropic effects of these genes indicated the potential involvement of coagulation factors, blood cell traits, liver function, and immunometabolic processes in VTE pathogenesis. In conclusion, our study unveils the valuable contribution of protein-coding variants in VTE etiology and sheds new light on its risk stratification.

Venous thromboembolism (VTE), consisting of deep vein thrombosis (DVT) and its complication, pulmonary embolism (PE), constitutes a major global public health challenge with high morbidity and mortality[1]. Given the steady increase in VTE incidence[2], it is crucial to elucidate the underlying risk factors associated with VTE to facilitate individualizing approaches for prevention[3].

VTE is a multifactorial disease involving interactions between acquired environmental and inherited genetic predispositions for thrombosis. Apart from recognized environmental factors such as immobilization, obesity, and advancing age[2], VTE is also strongly influenced by genetics, with twin-based heritability estimates over 40%[4,5]. However, previous genetic studies of VTE have focused primarily on common variants with higher minor allele frequencies (MAF > 1%). Such array-based (primarily noncoding variants) genome-wide association studies (GWAS)[6–10] meta-analyses have successfully identified 93

VTE risk loci of individually small effect sizes. By contrast, only a few rare variant studies of VTE with limited sample sizes and outcome events[11] exist to date. While it has revealed a substantial utility of rare variants, the identification of confidently associated genes remains scarce[11].

Compared to common variants, rare coding variants often tend to exhibit immediate biological effects[12], pinpoint causal genes[13], and guide subsequent experiments[14]. Rare monogenic variants, leading to the inactivation of natural anticoagulant proteins (*SERPINC1*[15], *PROC*[16], and *PROS1*[17]) or congenital fibrinogen disorders (*FGA*, *FGB*, and *FGG*)[18], have emerged as important contributors to autosomal dominant forms of thrombophilia[19]. And with widespread whole exome sequencing (WES), a novel gene-hunting approach has been provided for complex thrombotic disease[19].

To advance gene discovery for VTE beyond GWAS, we leveraged data from 349,038 UK Biobank (UKB) participants[20,21] and performed

[1]Department of Neurology and National Center for Neurological Disorders, Huashan Hospital, State Key Laboratory of Medical Neurobiology and MOE Frontiers Center for Brain Science, Shanghai Medical College, Fudan University, Shanghai, China. [2]Institute of Science and Technology for Brain-Inspired Intelligence, Fudan University, Shanghai, China. [3]Key Laboratory of Computational Neuroscience and Brain-Inspired Intelligence, Fudan University, Ministry of Education, Shanghai, China. [4]Department of Computer Science, University of Warwick, Coventry, UK. [5]These authors contributed equally: Xiao-Yu He, Bang-Sheng Wu, Liu Yang, Yu Guo, Yue-Ting Deng. ✉e-mail: wcheng@fudan.edu.cn; dong_qiang@fudan.edu.cn; jintai_yu@fudan.edu.cn

the WES analyses for VTE across the allele frequency spectrum. We showed that VTE was strongly influenced by 14 known and 4 novel protein-coding genes and corroborated them in the FinnGen cohort, all of which also exhibited significant associations with incident VTE risk in longitudinal prospective analyses. Notably, the identified genes, characterized by liver-specific expression, were enriched in coagulation cascade and platelet (PLT) activation. Furthermore, our findings also revealed the pleiotropic effects of the identified VTE genes, indicating the potential involvement of coagulation factors, blood cell traits, liver function, and immunometabolic markers in VTE development. Finally, our present study demonstrated that VTE is influenced by additive effects between rare coding variant burden and array-based common variant polygenic risk, and combining them may bolster the genetic risk stratification of VTE.

## Results

### Cohort characteristics
A total of 349,038 unrelated White British samples with "Caucasian" genetic ethnic grouping in the UKB were included in the WES analyses after sample quality control (QC) procedures, including 14,723 VTE cases and 334,315 controls (Supplementary Data 1). The median age at enrollment was 58.0 years, and 161,319 participants (53.8%) were women (more demographic and clinical characteristics of the study population by VTE case-control status were provided in Supplementary Data 2). 13,795,592 high-quality autosomal genetic variants remained in the exome sequencing data after variant QC. Among those, 13,695,488 variants were classified as rare (MAF < 1%), while 100,104 were classified as common (MAF ≥ 1%).

### Gene-level associations of rare coding variants with VTE
We first conducted a gene-level collapsing analysis to test associations of rare coding variants with VTE. To account for variations across genetic architectures, our framework encompassed 20,629 genes under 12 different models (see the "Methods" section and Supplementary Data 3). Within each gene, qualifying variants (QVs) containing either loss-of-function (LOF, stop gained, start lost, splice acceptor, splice donor, stop lost or frameshift) only, likely deleterious missense (Dmis, predicted to be deleterious by SIFT[22], LRT[23], PolyPhen2 HDIV, PolyPhen2 HVAR[24], and MutationTaster[25] consistently) only, or both LOF and Dmis were aggregated into distinct variant sets. These variant sets were created using QVs at four MAF thresholds ($<10^{-5}$, $<10^{-4}$, $<10^{-3}$, and $<10^{-2}$, Supplementary Data 3), and their impact on VTE was subsequently assessed. No substantial inflation was observed in any individual model, as indicated by genomic inflation factors consistently below 1.179 (Supplementary Fig. 1).

Applying a Bonferroni correction (corresponding to $P = 2.42 \times 10^{-6}$), we identified 30 significant associations implicating 6 VTE risk genes (*PHPT1*, *PROC*, *PROS1*, *SERPINC1*, *SRSF6*, and *STAB2*; Fig. 1A, Table 1 and Supplementary Data 4). Consistent with previous exome studies[10,11], four genes have significant deleterious effects on VTE risk (*PROC*, odds ratio (OR) = 1.07, 95% confidence interval (CI) = 1.05–1.10, $P = 1.22 \times 10^{-10}$; *PROS1*, OR = 1.08, 95% CI = 1.06–1.11, $P = 1.31 \times 10^{-9}$; *STAB2*, OR = 1.01, 95% CI = 1.01–1.02, $P = 5.98 \times 10^{-8}$; and *SERPINC1*, OR = 1.19, 95% CI = 1.11–1.28, $P = 8.89 \times 10^{-7}$). In addition, we uncovered *SRSF6* and *PHPT1* as novel VTE genetic risk genes. The strongest associations of *SRSF6* were observed in models for LOF-plus-Dmis burden with MAF $< 10^{-3}$ (OR = 1.04, 95% CI = 1.02–1.06, $P = 6.57 \times 10^{-7}$), while for *PHPT1* it was LOF burden with MAF $< 10^{-4}$ (OR = 1.08, 95% CI = 1.05–1.11, $P = 6.82 \times 10^{-7}$). The effect sizes of Dmis burden plus LOF or not in *SRSF6* were similar (OR = 1.04, Supplementary Data 4, 5), suggesting that detected significant signals were mainly driven by Dmis QVs in *SRSF6*. We also sought to confirm the exome-wide significant associations in FinnGen (release 8)[26]. Of the six identified genes, five replicated when searching for the most significant

variants mapped to each gene, while four were validated by our 'mBAT-combo' gene-based analysis ($P_{\text{Bonferroni}} < 0.05$, Table 1).

In subgroup analysis, the six VTE risk genes were nominally significant ($P < 0.05$) in both female and male groups and showed the same effect directions as in the full sample (Supplementary Fig. 2). Meanwhile, the male-specific analysis identified two more novel genes, *FNBP1L* and *APOBEC4* (Supplementary Fig. 3 and Supplementary Data 6). Furthermore, we have performed ancestry-specific and cross-ancestry meta-analyses in non-British White, Asian, Black, and Mixed populations, respectively (Supplementary Data 6). In ancestry-specific analysis, the associations with nominal significance ($P < 0.05$) showed the same effect directions as in the White British sample (Supplementary Fig. 4). As firm conclusions could not be drawn for some ethnic groups due to small sample sizes and low allele frequencies, cross-ancestry meta-analyses were performed. We found that the results were highly similar to those of only White British individuals (Supplementary Fig. 5), suggesting that the identified associations were not influenced by population stratification.

### Carrier frequency of 6 VTE-associated genes and VTE prevalence in variant carriers
We then quantified the carrier frequency of putatively pathogenic variants among six VTE-associated genes in unrelated UKB participants. Dmis variants were more prevalent in the population compared with LOF variants (Dmis 0.0112–0.9003%, LOF 0.0016–0.1581%), although the VTE prevalence in carriers was lower (Dmis 5.39–15.38%, LOF 6.83–54.55%; Fig. 1B, C, Supplementary Data 7). Furthermore, an inverse trend was observed between the frequency of variants and their phenotypic impact, indicating that alleles that contribute to higher VTE prevalence are generally held at lower frequencies in the population. Notably, with the lowest carrier frequency (0.0016%, 11 carriers), *SERPINC1* exhibited the largest effect size (Fig. 1D) and striking VTE prevalence (54.5%) in carriers.

### Burden heritability estimation for rare coding variants
Burden heritability regression (BHR)[27] was used to estimate the phenotypic variance (burden heritability) explained by the gene-wise burden of rare coding variants. In our analysis, the burden heritability for VTE ranged from 0.0073% to 0.1582% across different frequency bins and functional categories (Fig. 1E, Supplementary Data 8). Consistent with the previous report[27], ultra-rare LOF variants with MAF $< 10^{-5}$ explained a majority of the total heritability (0.1582%), but Dmis variants with higher MAF captured more variance than those with lower MAF. However, the aggregated burden heritability of the rare coding variants (0.5637%) is far less than the previously reported heritability based on common variants (13.2%)[10], which can be explained by the flattening hypothesis[28].

### Leave-one-variant-out (LOVO) analysis for 6 VTE-associated genes
To further assess the stability of our results and identify influential variants in genes associated with VTE, we performed the LOVO analysis. The associations in *PROC*, *PROS1*, *SERPINC1*, and *STAB2* remained robust when iteratively excluding each variant (Supplementary Fig. 6, Supplementary Data 9), suggesting these genes were identified due to a burden of multiple contributing rare variants.

Conversely, the significant associations of *PHPT1* and *SRSF6* with VTE were mainly driven by rs749387376 (p.Q84Gfs*98) and rs147863077 (p.R184H) respectively, although gene-level collapsing analysis excluding these two variants remained nominally significant (*PHPT1*, OR = 1.07, 95% CI = 1.02–1.12, $P = 5.48 \times 10^{-3}$; *SRSF6*, OR = 1.03, 95% CI = 1.01–1.06, $P = 1.28 \times 10^{-3}$; Supplementary Data 9). We also did additional Firth's bias-reduced logistic regression and confirmed that the most important variants within both genes were independently

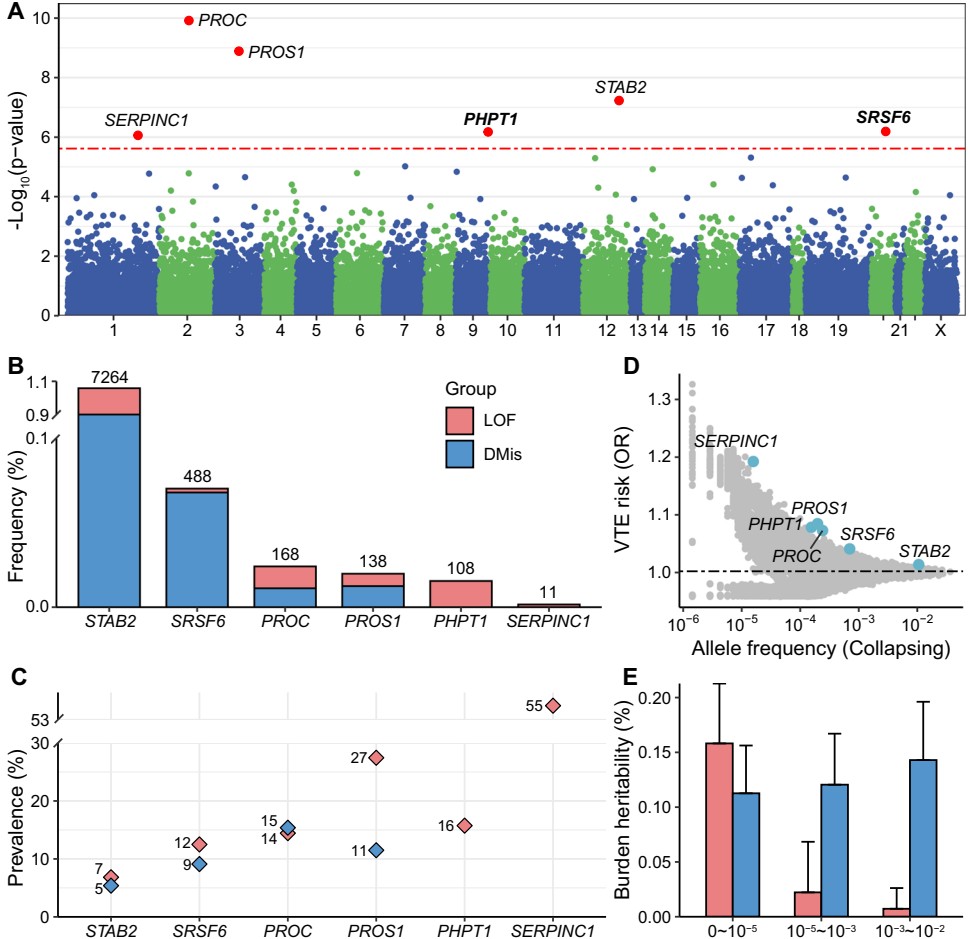

**Fig. 1 | Summary of gene-level associations with VTE among the unrelated White British population. A** Gene-based collapsing analysis results. Manhattan plot of results from gene-level association analysis in the 14,723 VTE cases and 334,315 control subjects. *P* values shown are two-sided, and Bonferroni correction was used. The red dotted horizontal line represents the Bonferroni significance threshold ($P = 2.42 \times 10^{-6}$). The colors on the plots show the delimitation of chromosomes. **B** Bar chart showing carrier frequencies for rare LOF and Dmis variants for six identified rare variant genes. The absolute number of carriers in the UK Biobank is shown above each bar. **C** VTE prevalence in LOF and Dmis QV carriers within genes associated with VTE in the collapsing analyses. **D** The distribution of ORs and allele frequencies for nominal significant collapsing genes (gray, *P* < 0.05). Genes that survived after multiple tests are labeled (blue). **E** Estimates of burden heritability across different frequency and functional categories (estimated using BHR), with colored bars showing burden heritability estimates and error bars showing the standard errors. The input summary statistics were obtained from a single variant association test for VTE (14,723 cases and 334,315 control subjects). **A**–**D** Display genes with the smallest *P* values achieved across all collapsing models. Source data are provided in the Source Data file, Supplementary Data 7 and 8.

associated with increased odds of VTE (rs749387376, OR = 6.52, 95% CI = 3.43–12.36, $P = 4.88 \times 10^{-9}$; rs147863077, OR = 2.52, 95% CI = 1.63–3.87, $P = 1.42 \times 10^{-5}$, Supplementary Data 10).

### Interplay between genome-wide polygenic risk score (PRS) and rare coding variants

Both common and rare alleles contribute to the risk of VTE, but their interaction has been insufficiently investigated due to limited datasets encompassing both common and rare variants. We calculated VTE genome-wide PRS ($PRS_{GW}$) in UKB. The $PRS_{GW}$ was derived from a VTE GWAS performed in an independent cohort, allowing us to explore its interplay with rare coding variants of substantial effect size in shaping risk for VTE[10].

First, analyses incorporating rare and common variants were performed based on $PRS_{GW}$ quintiles and rare variants' carrier status using logistic regression models. Relative to noncarriers in the average 40–60% $PRS_{GW}$ category, ORs ranged from 0.36 (95% CI = 0.33–0.39) to 3.92 (95% CI = 3.70–4.14) for noncarriers and from 0.54 (95% CI = 0.35–0.83) to 6.38 (95% CI = 5.50–7.39) for carriers in the lowest

and highest $PRS_{GW}$ quintile, respectively (Fig. 2, Supplementary Data 11).

Second, to probe potential interaction effects, we divided UKB individuals into quintiles based on their $PRS_{GW}$, and the prevalence of VTE was significantly higher in rare variant carriers than in noncarriers within most of the quintiles, demonstrating a significant interaction between $PRS_{GW}$ and rare variants (OR ranging from 1.36 to 1.63, *P* ranged from 0.0623 to $1.55 \times 10^{-11}$, Fig. 2, Supplementary Data 12). Although there was no interaction on the multiplicative scale (*P* ranged from 0.06 to 0.67 for multiplicative interaction, Supplementary Data 13), we observed a significant additive effect (relative excess risk due to interaction, RERI = 2.01, 95% CI = 1.22–2.79; synergy index, S = 1.58, 95% CI = 1.35–1.85; Supplementary Data 14). Our findings suggest a reciprocal enhancement between VTE $PRS_{GW}$ and rare variants in an additive manner, and one-third of the excess risk was attributable to the interaction between them (attributable proportion, AP = 0.31, 95% CI = 0.22–0.40).

We specifically explored the interaction between rare coding variants and factor V Leiden variant rs6025 (p.R534Q), a well-studied

**Table 1 | Significant associations with VTE in the gene-level collapsing analysis**

| Gene | Chr | Variant type; Frequency cutoff | Case count (n = 14,723) | Control count (n = 334,315) | OR | 95% CI | P-value | FinnGen VTE OR | FinnGen VTE P-value[a] | P-value (mBAT-combo)[b] |
|---|---|---|---|---|---|---|---|---|---|---|
| SERPINC1 | 1 | LOF; MAF<1e-4 | 6 | 5 | 1.19 | 1.11–1.28 | $8.89 \times 10^{-7}$ | 1.46 | 0.00139 | 0.0279 |
| PROC | 2 | LOF & Dmis; MAF < 0.001 | 25 | 143 | 1.07 | 1.05–1.10 | $1.22 \times 10^{-10}$ | 1.07 | $9.2 \times 10^{-7}$ | $9.09 \times 10^{-6}$ |
| PROS1 | 3 | LOF & Dmis; MAF < 1e-5 | 24 | 114 | 1.08 | 1.06–1.11 | $1.31 \times 10^{-9}$ | 1.28 | $9.65 \times 10^{-8}$ | $7.28 \times 10^{-5}$ |
| **PHPT1** | 9 | LOF; MAF < 0.001 | 17 | 91 | 1.08 | 1.05–1.11 | $6.82 \times 10^{-7}$ | 0.97 | 0.06 | 1 |
| STAB2 | 12 | LOF & Dmis; MAF < 0.01 | 412 | 6973 | 1.01 | 1.01–1.02 | $5.98 \times 10^{-8}$ | 1.53 | $3.39 \times 10^{-24}$ | $5.44 \times 10^{-14}$ |
| **SRSF6** | 20 | LOF & Dmis; MAF < 0.001 | 45 | 443 | 1.04 | 1.02–1.06 | $6.57 \times 10^{-7}$ | 1.26 | $3.35 \times 10^{-4}$ | 0.442 |

Case and control columns denote the count of QVs aggregated in the gene in the respective population. Genes highlighted in bold are not previously reported. P values shown are two-sided and unadjusted unless otherwise noted.
LOF loss-of-function, Dmis likely deleterious missense, MAF minor allele frequency.
[a]FinnGen VTE P-value after Bonferroni correction.
[b]P-value in gene-based association analysis using "mBAT-combo" method in GCTA software after Bonferroni correction.

**Fig. 2 | Additive effects between rare coding variants and PRS$_{GW}$.** ORs of VTE by PRS$_{GW}$ category and carrier status with reference to noncarrier 40–60% PRS$_{GW}$ category. Sample sizes of carriers and noncarriers by VTE case status and PRS$_{GW}$ category are indicated below the figure. ORs are plotted on a log scale, and error bars represent 95% CIs. Statistical differences in the prevalence between carriers and noncarriers were tested using a logistic regression analysis within each PRS$_{GW}$ quintile, and ORs were shown above the corresponding quintile group. P values shown are two-sided and unadjusted for multiple testing. *P values < 0.05, **P values < 0.01, ***P values < 0.001. Source data are provided in Supplementary Data 11 and 12.

high-risk common variant, for which only STAB2 showed additive effects (RERI = 1.17, AP = 0.30, S = 1.67, Supplementary Data 15).

**Single variant associations of common coding variants with VTE**
To further access the contribution of common coding variants to VTE risk without relying on imputation, we tested for the associations between common variants (MAF ≥ 1%) and VTE using an additive model in logistic regression by PLINK2[29]. We found 13 lead single nucleotide polymorphisms (SNPs) significantly associated with VTE after clumping (Fig. 3 and Table 2). Thirteen identified loci were mapped to 12 genes (including F5, PLCG2, SLC44A2, PROCR, KNG1, FGA, CYP4V2, KLKB1, ABO, VWF, CGN, and MAP3K2), among which 2 novel genes were not reported before (rs142913144 in CGN, OR = 0.76, 95% CI = 0.68–0.84, P = $6.83 \times 10^{-7}$; rs3732209 in MAP3K2, OR = 1.07, 95% CI = 1.05–1.10, P = $5.86 \times 10^{-8}$). We also queried the GWAS summary results for VTE in FinnGen to validate the associations that we identified, and 12 of 13 significant associations were replicated ($P_{Bonferroni}$ ranged from $1.47 \times 10^{-156}$ to $8.66 \times 10^{-3}$, Table 2). Subgroup analysis stratified by sex showed no significant sex heterogeneity of the identified loci (Supplementary Figs. 7, 8). Moreover, the identified associations were also significant in the cross-ancestry meta-analysis (Supplementary Figs. 9, 10) and were not influenced by population stratification as indicated by the high similarity with those of White British individuals (Supplementary Fig. 11). Furthermore, 10 out of 12 genes remained significant after inclusion of non-coding variants in the single variant analysis and linkage disequilibrium (LD)-based clumping process (Supplementary Fig. 12).

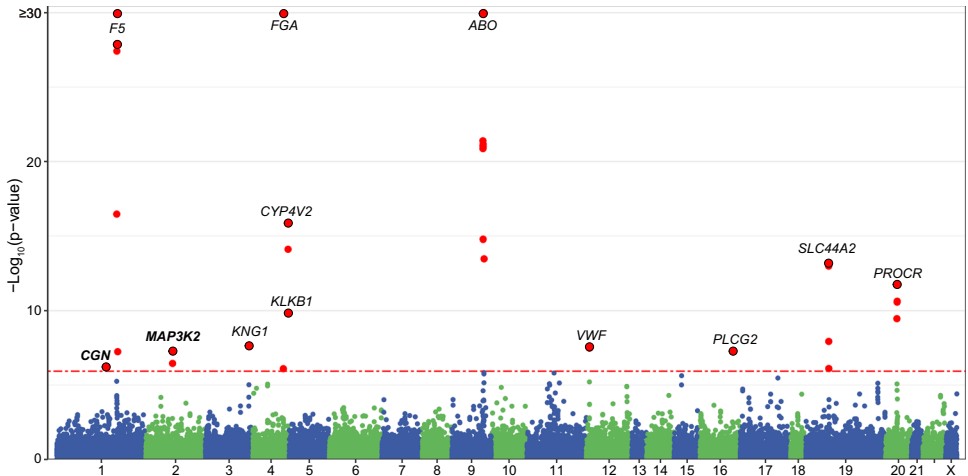

**Fig. 3 | Single variant associations between common variants and VTE.** Manhattan plot of results from single variant analysis for common variants in the 14,723 VTE cases and 334,315 control subjects. *P* values shown are two-sided and Bonferroni correction was used. The red dotted horizontal line represents the Bonferroni significance threshold ($P = 1.17 \times 10^{-6}$). The colors on the plots show the delimitation of chromosomes. Each lead SNP (red dots with black border) was annotated with its corresponding gene name. The *y*-axis is capped at 30. Source data are provided in Source Data file.

**Table 2 | Significant variant associations with VTE in the single-variant association analysis for common variants**

| rsID | Chr | AO | A1 | A1 freq | OR | 95% CI | *P*-value | Gene | Exonic function | AAChange | FinnGen VTE OR | FinnGen VTE *P*-value[a] |
|---|---|---|---|---|---|---|---|---|---|---|---|---|
| rs142913144 | 1 | G | A | 0.015 | 0.76 | 0.68–0.84 | $6.83 \times 10^{-7}$ | **CGN** | Nonsynonymous | p.R547K | 1.13 | 0.0898 |
| rs6016 | 1 | G | A | 0.272 | 0.86 | 0.83–0.88 | $1.2 \times 10^{-28}$ | F5 | Synonymous | | 0.87 | $2.41 \times 10^{-23}$ |
| rs6025 | 1 | C | T | 0.023 | 2.38 | 2.25–2.52 | $1.99 \times 10^{-202}$ | F5 | Nonsynonymous | p.R534Q | 2.35 | $1.47 \times 10^{-156}$ |
| rs3732209 | 2 | A | G | 0.316 | 1.07 | 1.05–1.10 | $5.86 \times 10^{-8}$ | **MAP3K2** | Synonymous | | 1.06 | $1.83 \times 10^{-4}$ |
| rs1656922 | 3 | C | T | 0.489 | 0.94 | 0.91–0.96 | $2.58 \times 10^{-8}$ | KNG1 | Nonsynonymous | p.M178T | 0.96 | 0.00866 |
| rs6050 | 4 | T | C | 0.255 | 1.21 | 1.18–1.24 | $2.11 \times 10^{-48}$ | FGA | Nonsynonymous | p.T331A | 1.2 | $3.38 \times 10^{-52}$ |
| rs3736455 | 4 | G | T | 0.332 | 0.9 | 0.88–0.92 | $1.35 \times 10^{-16}$ | CYP4V2 | Synonymous | | 0.89 | $6.38 \times 10^{-22}$ |
| rs925453 | 4 | C | T | 0.313 | 0.92 | 0.90–0.94 | $1.59 \times 10^{-10}$ | KLKB1 | Nonsynonymous | p.W503R | 0.9 | $5.41 \times 10^{-17}$ |
| rs8176719 | 9 | T | TC | 0.338 | 1.31 | 1.28–1.34 | $6.94 \times 10^{-106}$ | ABO | Frameshift Substitution | p.T87Dfs*107 | 1.27 | $1.57 \times 10^{-98}$ |
| rs1063856 | 12 | T | C | 0.376 | 1.07 | 1.04–1.10 | $3.02 \times 10^{-8}$ | VWF | Nonsynonymous | p.T789A | 1.06 | $1.87 \times 10^{-5}$ |
| rs1143686 | 16 | A | G | 0.320 | 1.07 | 1.04–1.10 | $5.87 \times 10^{-8}$ | PLCG2 | Synonymous | | 1.04 | 0.00481 |
| rs2288904 | 19 | G | A | 0.223 | 0.9 | 0.87–0.92 | $6.86 \times 10^{-14}$ | SLC44A2 | Nonsynonymous | p.Q154R | 0.89 | $2.37 \times 10^{-14}$ |
| rs867186 | 20 | A | G | 0.088 | 1.15 | 1.11–1.20 | $1.87 \times 10^{-12}$ | PROCR | Nonsynonymous | p.S219G | 1.13 | $1.08 \times 10^{-13}$ |

Genes highlighted in bold are not previously reported. *P* values shown are two-sided and unadjusted unless otherwise noted.
a FinnGen VTE *P*-value after Bonferroni correction.

## Association between identified coding variant genes and incident VTE

Time-to-event analysis was performed to investigate the association between identified genes, lead SNPs, and incident VTE risk. Among 300,523 participants without VTE at baseline, 6920 incident VTE cases were identified. During a median (inter-quartile range) follow-up of 9.23 (7.07–11.13) years, 5 of 6 genes and all lead SNPs identified from WES analysis were significantly associated with incident VTE ($P_{Bonferroni}$ ranged from $2.88 \times 10^{-66}$ to 0.021, Fig. 4A). In comparison to noncarriers, rare variant gene carriers exhibited hazard ratios (HRs) ranging from 1.37 (*STAB2*, 95% CI = 1.20–1.57, $P_{Bonferroni} = 2.72 \times 10^{-5}$) to 6.65 (*PROS1*, 95% CI = 4.01–11.04, $P_{Bonferroni} = 1.39 \times 10^{-12}$), while common lead SNP carriers showed moderate HRs ranging from 0.71 (rs142913144, 95% CI = 0.60–0.84, $P_{Bonferroni} = 5.56 \times 10^{-4}$) to 2.08 (rs6025, 95% CI = 1.91–2.26, $P_{Bonferroni} = 2.88 \times 10^{-66}$) (Fig. 4A, Supplementary Data 16). Notably, the effect sizes of rare variants were

substantially larger than those of common variants, indicating a more pronounced impact.

Then we estimated the aggregated effect of rare and common coding variants on VTE risk. As shown in Fig. 4B, there was a significant difference in incident VTE between rare coding variant carriers and noncarriers over the entire follow-up period (HR = 1.58, 95% CI = 1.40–1.79, $P = 8.64 \times 10^{-13}$). In addition, a significant difference in incident VTE risk also existed among categorical low and high common variant PRS strata (HR = 1.66, 95% CI = 1.59–1.74, $P = 9.14 \times 10^{-97}$).

## Biological function and pleiotropy of VTE-associated genes

To further elucidate the biological relevance of the WES analysis results, we conducted gene-set analyses with hypergeometric tests for the 18 VTE-associated genes using the GENE2FUNC function in FUMA[30]. We found most of the gene ontology (GO) biological process terms were significantly associated with hemostasis ($P = 4.12 \times 10^{-15}$),

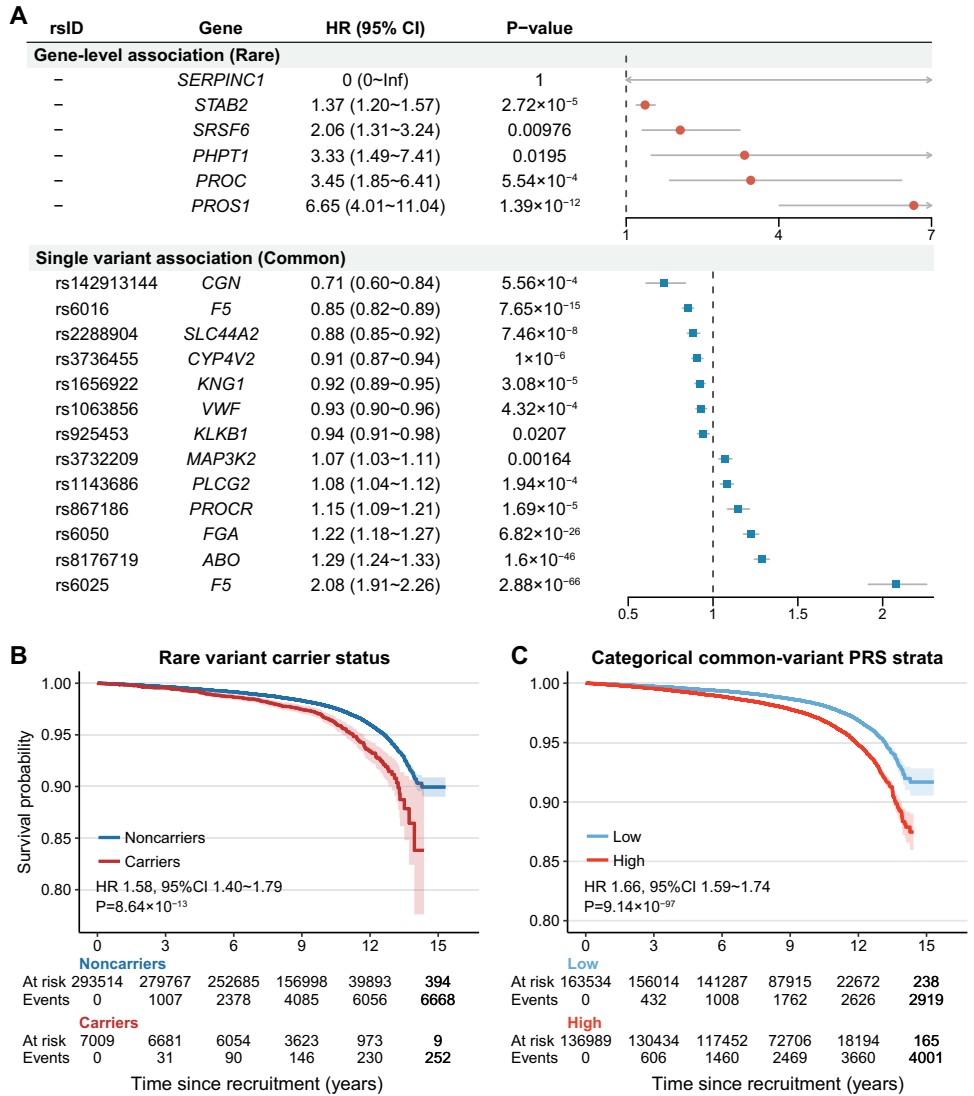

**Fig. 4 | Longitudinal association between identified genes or coding lead SNPs and incident VTE. A** Forest plot showing HR and 95%CI for the association between identified genes (from gene-level collapsing analysis) or coding lead SNPs (from single variant analysis) and incident VTE. HR and 95% CI were calculated through CPH regression in participants without VTE at baseline (6920 incident VTE cases and 293,603 controls) and adjusted for age, sex, and the first 10 PCs. Bonferroni correction was applied, respectively, for the number of significant genes or lead SNPs in that particular analysis, and *P* values after Bonferroni correction were provided. Kaplan–Meier survival curves of VTE are plotted according to **B** rare variant carrier status and **C** categorical low and high common-variant PRS strata (constructed using coding lead SNPs). Follow-up time since recruitment was truncated at 15 years. HR, 95% CI, and *P* value provided were calculated through CPH regression in participants without VTE at baseline (6920 incident VTE cases and 293,603 controls) and adjusted for age, sex, and the first 10 PCs. All *P* values shown are two-sided. Source data are provided in Supplementary Data 16.

and in terms of cellular components, PLT alpha granule lumen ($P = 5.04 \times 10^{-9}$) was predominantly involved (Supplementary Data 17). Meanwhile, we found an overrepresentation of genes that are highly and specifically expressed in the liver, which is a key center for synthesizing coagulation factors and regulatory proteins and plays a key role in hemostatic and thrombotic regulation ($P = 2.35 \times 10^{-8}$, Supplementary Data 18). Furthermore, leveraging the summary data from the GWAS catalog, we revealed significant enrichment of VTE-related genes in diseases such as thrombosis and ischemic stroke as well as in hemostatic factors and hematological phenotypes such as D-dimer levels and von Willebrand factor (vWF) levels (Supplementary Data 17).

To investigate the pleiotropic effects of VTE-associated genes for a better understanding of VTE pathogenesis, we explored the associations between VTE-associated genes and 35 curated phenotypes, including blood cells, immunometabolic markers, liver function indicators, and coagulation factors (Supplementary Data 19). As expected, a large proportion of these phenotypes showed significant associations with the genes we discovered. Moreover, the significant associations mainly fell within common variant signals, in particular, *ABO* (24 of 35 traits) and *SLC44A2* (6 of 35 traits) (Supplementary Fig. 13, Supplementary Data 20). Within rare variant signals, we only observed significant associations of *STAB2* with insulin-like growth factor 1 ($P = 5.96 \times 10^{-10}$), vWF ($P = 1.69 \times 10^{-19}$), and gamma-glutamyltransferase ($P = 2.96 \times 10^{-6}$). Besides, *SRSF6* showed a nominal association with PLT count ($P = 0.0028$) and glycated hemoglobin ($P = 0.0478$). In line with previous studies[10], our findings reinforce the central role of blood cell traits and coagulation factors in VTE pathogenesis and demonstrate the potential involvement of liver function and immunometabolic processes in the occurrence of VTE.

## Discussion

Leveraging exome sequencing data from 349,038 participants, we performed a WES analysis of VTE. We discovered 14 known and 4 novel genes associated with VTE risk, 16 of which were replicated in the FinnGen cohort. Each of these significant genes exhibited the potential to identify individuals at risk (that is, incident cases), as demonstrated by the fact that carriers of deleterious variants were more likely to develop VTE, particularly those with low allele frequencies. Functional annotation supported the enrichment of these genes in pathways related to coagulation or PLT function and highlighted the involvement of the liver in the biological process of VTE. The pleiotropic effects of these genes on coagulation factors, blood cell traits, liver function, and immunometabolic markers were also uncovered. Moreover, and importantly, this study provided evidence that rare coding variants affected VTE additively to common $PRS_{GW}$, suggesting that the genetic assessment of VTE could be further improved by combining $PRS_{GW}$ and rare coding variant burden.

Among the significant VTE-associated loci reported in previous studies, our study provides clear replication for the widely known genetic mutations, including the anticoagulant genes *PROC*, *PROS1*, *SERPINC1*, and *PROCR*[11,31,32]; the procoagulant genes *F5*, *KLKB1*, and *KNG1*[33,34]; the fibrinolytic disorder-related gene *FGA*[33,35]; and the *ABO*, *STAB2*, and *VWF* genes that act on von Willebrand factors and indirectly influence coagulation activity[10,11,32,36]. The recently discovered variations within *SLC44A2*, *CYP4V2*, and *PLCG2*[6,8,37] were also verified in our findings, albeit their exact roles involved in VTE pathogenesis remain ambiguous and need to be further explored. Despite this, we found the relationship between *PLCG2* and functions converging on the coagulation cascade or PLT function, in line with recent reports from animal studies[38]. Our study also extends former findings via the discovery of four additional novel candidate genes (*SRSF6*, *MAP3K2*, *CGN*, and *PHPT1*), which may provide valuable insights into genetic loci not previously suspected to play a role in VTE. Importantly, two of these four genetic associations could be validated in the FinnGen cohort. From the perspective of clinical application, each of the significant genes we discovered exhibited the potential to predict future VTE events. Carriers of mutations in our newly discovered genes, *SRSF6* and *PHPT1*, have more than 2-fold higher risk of developing VTE than noncarriers, which reinforces the reliability and credibility of our analyses.

Notably, effect sizes for identified genes from collapsing analysis are lower than expected from previous studies. Considering that HR estimates from subsequent survival analysis were larger and more consistent with previous reports, we hypothesize that the comparatively smaller effect sizes observed during the initial gene discovery phase were primarily due to the usage of saddle point-approximation-corrected logistic mixed-model approach implemented in the SAIGE-GENE+ software, which might yield slightly conservative effect estimates, particularly when assessing significance for binary traits with imbalanced case-control ratios[39].

Regarding the newly identified *SRSF6* gene, its encoded protein belongs to the splicing factor SR family, which is involved in mRNA splicing and may be responsible for the determination of alternative splicing[40]. The tissue factor, the initiator of the coagulation process, may lead to thrombus growth by creating a surface that both initiates and propagates coagulation through a variant generated by alternative splicing to bind to the edge of the thrombus[41]. This supports a possible relationship between the *SRSF6* gene and VTE formation. Moreover, past studies lent support to the link between *SRSF6* and mean PLT volume[42]. Our PheWAS analyses showed a nominally significant association between *SRSF6* and PLT count. Interestingly, both the enlarged PLTs, as measured by mean PLT volume, and the PLT count were reported to be associated with VTE events[32,43–45]. Collectively, our study provides initial insights into the biological pathways by which the *SRSF6* gene engages in VTE generation.

As for *MAP3K2*, its association with protein C levels has been disclosed[46]. Protein C is a molecule with potent anticoagulation properties. Once activated, it inactivates Factor Va and Factor VIIIa, thereby inhibiting thrombin generation. The rate of VTE is three to seven times higher in people with protein C deficiency relative to those with a normal range of protein C[47,48], and 2–3% of VTE cases in the population may be attributed to protein C deficiency[48]. The relationship we found between *MAP3K2* and immunometabolic measures, white blood cell traits, and liver function has been less studied before, so more research is needed to elucidate the underlying mechanisms. *PHPT1*, a gene implicated in the modulation of phosphorylation activity, has previously been established as an oncogene in a variety of tumorigenic processes[49]. Yet, the putative role of *PHPT1* in VTE remains largely unknown. Relevant in-depth studies should be carried out in the future. In addition, although most of the novel associations we observed had relatively small effect sizes (ORs ranging from 1.04 to 1.08), we were able to identify a common variant with larger estimated effects in the *CGN* gene (OR = 0.75). In the present study, we confirmed multiple genetic-phenotypic associations previously described of the *CGN* gene with lipid metabolism and immunity[50,51], and the latter is related to VTE risk in numerous articles[52–55]. Our findings suggest a potential involvement of these novel genes in VTE, and future independent and diverse studies will be required to verify our hypothesis.

Current therapeutic strategies are primarily targeted at the coagulation cascade. While the safety of anticoagulation therapy has improved in recent years, bleeding is still a life-threatening off-target outcome[56]. Tablets that suppress PLT activation, for instance, aspirin, may exert beneficial effects in preventing VTE, despite previous studies and trials on the use of anticoagulants in conjunction with aspirin yielding discordant results[32,57]. Innovative approaches are thus urgently needed for preventing thrombosis whilst minimizing bleeding[56]. In this scenario, our newfound gene, such as *SRSF6*, may have the potential to break the inexorable bond between antithrombotic therapy and bleeding risk and, in turn, serve as a candidate drug target. However, there is plenty of arduous validation work to be done before it can actually move into clinical implementation. Even so, our research work offers a new direction to guide future mechanistic investigations and constitutes a valuable resource for thrombosis researchers and the discovery of new VTE therapeutic targets.

Although an array of common variants has been uncovered by prior well-powered VTE GWAS[6–10], they may not fully capture the heritable risk for this complicated illness. Our study demonstrated this by revealing that rare coding variants affected VTE additively to common variant-based PRS. While the interaction between rare variants and $PRS_{GW}$ has been reported in other common phenotypes[58–60], it is, to our best knowledge, the first time that such an interplay has been unveiled in VTE. Considering $PRS_{GW}$ and carrier status jointly, the absolute risk is consistently lower for noncarriers than for carriers in the same $PRS_{GW}$ quintile, which indicates that the genetic assessment of VTE can be further improved by combining $PRS_{GW}$ and rare coding variant burden. This has great clinical implications, for example, to inform decisions on VTE screening, where rare variant holders with high $PRS_{GW}$ may gain benefit from earlier and more frequent screening. Our data suggest that extending the current $PRS_{GW}$ panel to include testing for rare variants may contribute to a more accurate and comprehensive estimate of the genetic risk for VTE and may be warranted.

The main advantage of this genetic discovery effort lies in the large sample size. Compared to previous GWAS and WES analyses on VTE, this study enhanced statistical power and strengthened the ability to uncover novel genes. The estimation of burden heritability and the interplay between rare variants and known $PRS_{GW}$ further refine the understanding of genetic architecture. Our findings should be interpreted within the context of the limitations. First, this study was focused on coding variants, and we did not pay much attention to the

contribution of non-coding variants. Second, as non-White British populations were under-represented in UKB, future studies with even larger sample sizes should be warranted in other ancestries. Third, we were unable to identify whether VTE cases were provoked or unprovoked due to insufficient information in UKB. Furthermore, GWAS summary data from FinnGen is not ideal enough for perfectly replicating our results from exome-sequencing data. However, it can still provide evidence to support our results. This may limit the power to analyze diseases and may lead to decreased burden heritability estimates to some extent[27].

In conclusion, large-scale sequencing has enabled a comprehensive dissection of the genetic factors predisposing to VTE. We have both confirmed many known gene–disease associations and identified novel genetic variants. Some of the new genes may contribute to VTE via well-characterized coagulation pathways or by influencing hematological traits, particularly PLT function, while others provide new evidence beyond established or currently hypothesized pathways to thrombosis. Furthermore, integrating genetic information on the rare variant burden with known genome-wide PRS may facilitate the differentiation between low- and high-risk patients.

## Methods

### Study population

UKB is a large prospective cohort of ~500,000 individuals aged 40–69 years at baseline between 2006 and 2010[20]. Only ~450,000 participants who were self-reported as White British ethnicity with complete exome sequencing and VTE diagnostic information were included. Ethical approval was obtained from the National Health Service National Research Ethics Service and all participants gave informed consent through electronic signatures. This study was performed based on application number 19542.

The VTE diagnosis in this project was based on self-report at baseline or electronic health records (EHR). We used the International Classification of Diseases system codes 10th revision (ICD-10) codes (I80.1, I80.2, I82.2, I26.0, and I26.9) and Office of Population and Censuses and Survey 4th Revision Procedures Codes (OPCS-4, L79.1 and L90.2) to identify VTE cases, which presented as a primary or secondary diagnosis in the hospital inpatient records or an underlying cause of death in the death register (Supplementary Data 1). Furthermore, we excluded individuals with a diagnosis of superficial or unclear site of thrombophlebitis (I80.0, I80.3, I80.8, I80.9), portal vein thrombosis (I81), BuddChiari syndrome (I82.0), and known coagulation defects (D68) to minimize heterogeneity and bias[9,61].

### Genotyping, QC, and variant annotation

All participants' exome sequencing was conducted in the Regeneron Genetics Center (RGC), with detailed methods described in Supplementary Note 1[21]. We used the OQFE WES pVCF files provided by the UKB (https://biobank.ctsu.ox.ac.uk/showcase/label.cgi?id=170), aligned to the human reference genome GRCh38[62], and performed similar extensive QC as in previous study[63]. After splitting multi-allelic sites into bi-allelic sites, low-quality and extreme outlier genotypes were removed using Hail. Moreover, variants in low-complexity regions, with call rates < 90% and departure from the Hardy–Weinberg equilibrium ($p < 1 \times 10^{-15}$), were excluded. For chromosome X, pseudoautosomal (PAR1 and PAR2) regions were further excluded in both sexes, and heterozygous genotypes in males were set to missing[21]. For sample QC, we excluded participants withdrawn from the UKB, duplicates, gender-mismatched, high-related samples (second-degree or closer relatives, kinship coefficient threshold at 0.0884, see Supplementary Note 2), and whose Ti/Tv, Het/Hom, SNV/indel and number of singletons deviating from mean ± 8 SD. Finally, we restricted our main analysis to White British (filed 21000) with 'Caucasian' genetic ethnic grouping[64], and used high-quality variants to calculate the principal components (PCs) within ancestry. Of 349,038 unrelated

participants, we identified 14,723 VTE cases and 334,315 controls according to the definition by Klarin et al.[61].

We used the SnpEff tool[65] to annotate variants and ascertained the most severe consequence for each gene transcript. For the analysis, we considered only LOF (stop gained, start lost, splice acceptor, splice donor, stop lost or frameshift) and Dmis variants (predicted to be deleterious by all five in silico algorithms, namely SIFT[22], LRT[23], PolyPhen2 HDIV, PolyPhen2 HVAR[24], and MutationTaster[25]).

### Rare variants collapsing analysis

For rare variants with MAF < 0.01, a gene-level collapsing analysis was performed. We included 12 models to qualify variant criteria for 20,629 genes, which vary in terms of MAF ($<10^{-5}$, $<10^{-4}$, $<10^{-3}$, and $<10^{-2}$), and predicted consequence (LOF, Dmis, and LOF-plus-Dmis).

We fitted a logistic mixed-effects model to identify genes relevant to VTE in unrelated White British individuals. Covariates, including sex, age, and top 10 PCs, were adjusted as fixed effects in all association analyses to minimize confounding and potential population stratification. The sparse genetic relationship matrix constructed using the high-quality variants with the recommended relative coefficient cutoff of 0.05 was included as a random effect for the variance ratio estimation. The effect sizes and $P$ values were calculated for each collapsing association using tests such as SKAT-O[66], burden[67], and SKAT[68] implemented in SAIGE-GENE+[69], with SKAT-O reported in the main article. We also performed subgroup analysis stratified by sex (female and male) to investigate the sex heterogeneity of the identified genes. All performed tests were two-sided, and the statistical significance threshold was determined to be $2.42 \times 10^{-6}$ by using a Bonferroni correction for 20,629 genes.

Upon identifying the significant risk genes for VTE, carriers of QVs included in the most significant models per gene were denoted as rare damaging coding variant carriers. We further calculated the disease prevalence of the susceptibility genes for VTE, which represents the percentage of VTE cases among carriers with these rare variants.

### Burden heritability estimation

To estimate the phenotypic variance (burden heritability) explained by the gene-wise burden of rare coding variants, we used BHR[27]. A single variant association test for VTE was performed to get variant-level association summary statistics, which employed Firth's bias-reduced logistic regression utilizing SAIGE-GENE+ with the same covariates set up as in collapsing analysis.

After inputting variant-level association summary statistics and allele frequencies, burden heritability could be quantified from the slope of burden test statistics regressed on burden scores[27]. Following the default settings of BHR with the provided baseline model, we focused primarily on LOF and Dmis variants and estimated burden heritability separately for three different MAF categories: $[0, 10^{-5})$, $[10^{-5}, 10^{-3})$, and $[10^{-3}, 0.01)$.

### LOVO analysis

LOVO analyses were performed using our significant gene-level findings to identify individual variants contributing to aggregated test statistics. We iteratively excluded each variant at a time from the most significant models and reran the association test to examine how this affected statistical significance. The most important variant within each gene was defined as the variant that was removed to achieve a maximum association test $P$-value.

### Interplay of genome-wide PRS and rare variants on VTE risk

Combined effect analyses focused on the carrier status of rare damaging coding variants and $PRS_{GW}$ constructed using recently published GWAS[10] (see $PRS_{GW}$ derivation details in Supplementary Note 3). The $PRS_{GW}$ was subsequently quintiled as categorical variables using cutoffs determined by its distributions in controls[60].

First, we established an indicator variable based on $PRS_{GW}$ quintile and rare variants carrier status, with noncarriers in the 40–60% $PRS_{GW}$ category as the reference. Logistic regression models, added covariates including age, sex, and 10 PCs, were used to calculate ORs for each of these categories. Second, stratified analyses by $PRS_{GW}$ quintiles were undertaken. We compared the prevalence of VTE between rare variants carriers and noncarriers within each quintile. Statistical difference was tested using logistic regression models with the same covariates adjusted, and the ORs, corresponding 95% CIs, and *P* values for each quintile were reported (Supplementary Data 12).

We further explored the interaction manner between $PRS_{GW}$ and rare variants carrier status for each gene and all six rare variant genes in aggregate. Additive effects were measured as the RERI, S, and AP due to interaction using epiR package, with dichotomous categories for standardized $PRS_{GW}$ employed: low (<0) and high (>0) risk. Multiplicative interactions were measured by adding an interaction term of continuous $PRS_{GW}$ and rare variants carrier status, and then statistical tests for significant interaction effects were performed. The two-sided $P < 0.05$ were considered statistically significant.

### Single variant analysis for common variants

A single variant analysis was performed among unrelated White British population using PLINK2[29] for common exonic SNPs (MAF ≥ 1%), with an additive genetic model employed in logistic regression. Age, sex, and the first 10 PCs were adjusted, and the significance threshold was set to $1.17 \times 10^{-6}$ (Bonferroni correction for 42,911 exonic coding SNPs). After that, significant SNPs were selected to identify the lead SNPs within each risk locus, which were denoted as the SNPs with the lowest p-value when multiple SNPs were observed to be in strong LD ($r^2 > 0.01$) within a 5000-kb window. Risk loci were defined as regions of ±1 Mb around each lead SNP. Sex-stratified sensitivity analyses were conducted to investigate the sex heterogeneity of the identified loci. Furthermore, we also conducted a single variant analysis using common SNPs in both exonic and non-exonic regions to test whether the identified coding variants were still independently significant after LD-clumping.

### Multi-ancestry analysis

In order to improve the generalizability of our results, we further performed a gene-level collapsing analysis in non-British White (25,671 samples, 936 VTE cases), Asian (8558 samples, 194 VTE cases), Black (6628 samples, 251 VTE cases), and Mixed (5961 samples, 188 VTE cases) population, separately. These ethnicities were defined using the self-reported ethnic background (Field 21000 in UKB). After that, the summary statistics of five ethnic groups (British White, non-British White, Asian, Black, and Mixed) were meta-analyzed together using METAL[70]. Similarly, we conducted a single-variant association analysis in each ethnic group, respectively, and the summary statistics were meta-analyzed using METAL[70]. We compared the effect sizes and significances of the results between cross-ancestry and White British population using Pearson correlations.

### Time-to-event validation

To compare the incidence rate of VTE in different carrier statuses of each gene, Cox proportional hazards (CPH) regression with the same covariates described above was performed, and HRs were reported. Time zero was the date of recruitment to UKB, and follow-up time was subsequently calculated as years from it to the date of first diagnosis, death, or the final date with accessible information from hospital admission, whichever came first. We excluded 48,515 participants with a VTE diagnosis before recruitment or without follow-up, and the remaining 300,523 participants were included in the longitudinal analysis.

Within each significant gene in collapsing analysis, QVs contained in the model of strongest association with VTE were aggregated into a single variable to represent that gene. We also collapsed all significant genes, distinguishing carriers from noncarriers, to study the combined effects of rare coding variants. For significant common variants, in addition to variant-level analysis, we calculated a PRS constructed using the LD-clumped lead SNPs by PLINK2 and divided the participants into 2 groups: low (standardized PRS < 0) and high (standardized PRS > 0) risk to represent combined effects of common coding variants. Then Kaplan–Meier analyses and the CPH regression were conducted to investigate whether the survival probability of incident VTE differs substantially.

### Functional mapping and annotation

To functionally annotate the biological relevance in our WES findings across the allele frequency spectrum, gene-based analysis was performed using MAGMA[71] to query biological annotation and pathway databases (including GO[72], KEGG[73], Reactome[74], and GWAS catalog) through functional mapping and annotation of genetic associations (FUMA) platform[30]. Additionally, FUMA was also used to perform tissue enrichment analysis in 30 broad tissue types based on data from the GTEx database (version 8)[75]. *P* values were adjusted for multiple testing by the Bonferroni approach.

### External replication in FinnGen

For external replication of our identified signals in an independent sample cohort, we used summary statistics from the FinnGen Consortium online results (version 8)[26], which included 17,048 VTE cases and 325,451 controls.

FinnGen study contained genotype and national health registry data of Finland citizens, which was collected from different Finnish biobanks and digital health care data since 2017. Genotyping was done using various Illumina and Affymetrix GWAS arrays and further imputed to 20 million variants by the Beagle software (v.4.1) based on population-specific SISu v4.0 imputation reference panel, whole genome sequences of 8554 Finnish individuals. We focused on the disease endpoint, "Venous thromboembolism". The genetic association tests were applied using Regenie for variants with a minimum allele count of 5, with the Firth test for variants with an initial *P*-value of <0.01. The summary statistics were publicly available online (see data availability). For gene-level collapsing analysis validation, we searched for variants with the strongest FinnGen VTE associations mapped to significant genes we found. We further performed a gene-based association analysis using the summary statistics as input in GCTA software, with "mBAT-combo" command combining multi-SNP statistics effectively through a Cauchy combination method[76]. For single variant association validation, we searched for our lead SNPs directly. Both approaches used Bonferroni correction for the number of significant genes or lead SNPs in that particular analysis.

### Phenome-wide association analysis (PheWAS)

To further elucidate the potential pathophysiological mechanisms by which risk genes contribute to VTE, we conducted PheWAS. This analysis focused on a curated list of 35 blood traits under five categories in UKB, which encompassed six coagulation factors (from UKB plasma proteins data measured using the Olink Explore 1536 platform in a subset of ~50,000 individuals), 11 immunometabolic markers, 6 liver function indicators, 4 red blood cell traits, and 8 white blood cell traits. Blood trait values outside four standard deviations from the mean were considered outliers and excluded from the analyses. Detailed information about these traits was listed in Supplementary Data 19. Associations of PheWAS traits with rare variant genes were analyzed using gene-level linear mixed models in SAIGE-GENE+, while linear models were applied for common lead SNPs. All models were adjusted for age, sex, and the first 10 PCs. The significance threshold was set at $P = 7.51 \times 10^{-5}$ (Bonferroni correction for 6 risk genes, 13 associated lead SNPs, and 35 traits).

## Reporting summary

Further information on research design is available in the Nature Portfolio Reporting Summary linked to this article.

## Data availability

Individual-level data from the UKB samples are available through UKB under application number 19542. FinnGen GWAS summary statistics are publicly accessible (http://r8.FinnGen.fi). The gene-level and single variant association summary statistics generated in this study have been made accessible through https://doi.org/10.6084/m9.figshare.25433899.

## Code availability

The following software and packages were used for data analysis: FUMA v.1.3.8 (https://fuma.ctglab.nl/), SnpEff v.5.1 (https://pcingola.github.io/SnpEff/), SAIGE-GENE+ v.1.1.6.2 (https://github.com/saigegit/SAIGE), BHR v.0.1.0 (https://github.com/ajaynadig/bhr), PLINK v.2.0 (https://www.cog-genomics.org/plink/), mBAT-combo in GCTA v.1.94.1 (https://yanglab.westlake.edu.cn/software/gcta/#mBAT-combo), METAL v.2011-03-25 (http://csg.sph.umich.edu/abecasis/Metal/), and R v.4.2.0 (https://www.r-project.org/).

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

## Acknowledgements

We gratefully thank all the participants and professionals for collecting and preparing data in the UKB and FinnGen study. We sincerely appreciate for support of the Medical Research Data Center of Fudan University. J.T. Yu was supported by grants from the Science and Technology Innovation 2030 Major Projects (2022ZD0211600), National Natural Science Foundation of China (82071201, 81971032, 92249305), Shanghai Municipal Science and Technology Major Project (No. 2018SHZDZX01), Research Start-up Fund of Huashan Hospital (2022QD002), Excellence 2025 Talent Cultivation Program at Fudan University (3030277001), Shanghai Talent Development Funding for The Project (2019074), and ZHANGJIANG LAB, Tianqiao and Chrissy Chen Institute, and the State Key Laboratory of Neurobiology and Frontiers Center for Brain Science of Ministry of Education, Fudan University. W. Cheng was supported by grants from the National Natural Sciences Foundation of China (no. 82071997) and the Shanghai Rising-Star Program (no. 21QA1408700). J.F. Feng was supported by the National Key R&D Program of China (No. 2018YFC1312904 and No. 2019YFA0709502), the Shanghai Municipal Science and Technology Major Project (No. 2018SHZDZX01), the 111 Project (No. B18015), Shanghai Center for Brain Science and Brain-Inspired Technology and Zhangjiang Lab.

## Author contributions

J.T. Yu, Q. Dong, W. Cheng, and J.F. Feng designed the study. X.Y. He and Y. Guo conducted the main analyses and drafted the manuscript. L. Yang, B.S. Wu, Y.T. Deng, Z.Y. Li, C.J. Fei, Y.J. Ge, J.J. Kang, and W.S. Liu contributed to data collection and analyses. J.T. Yu, Q. Dong, W. Cheng, and J.F. Feng critically revised the manuscript. All authors reviewed and approved the final version.

## Competing interests

The authors declare no competing interests.
