## [Peer Review File · Nature Communications]

Genetic associations of protein-coding variants in venous thromboembolismREVIEWER COMMENTS

Reviewer #1 (Remarks to the Author):

This is a nice manuscript that uses the resources of the UK Biobank to conduct a rare genetic variant analyses of venous thromboembolic disease. This allows the comparison of a large group of cases ~14,000 to an even larger group of controls >300,000. The authors employ up to date analysis pipelines and perform several ancillary studies to complement the results of their gene collapsing analyses. In addition to the collapsing analysis, I particularly enjoyed the addition of rare variants to the common variant polygenic risk score which demonstrated additive interaction with common variants. This study can be improved through a couple additional analyses, clarification and re-writing of the majority of the discussion section as outlined below.

Major points

1. In order to better understand the VTE cases and controls the authors must better explain the criteria they used to define VTE cases. I don't think this is anywhere in the manuscript. Were ICD9/10 codes used? How do the authors account for trauma, pregnancy, central line or cancer related VTE?
2. No analyses are provided of the X-chromosome. This is not acceptable. Many coagulation genes are encoded for on the X-chromosome. Please use the UKBB and FinnGenn X-chromosome sequencing data.
3. Although the authors define the 12 different collapsing models used for their primary analysis, this should be briefly explained in the results section, a simple table with allele frequency, variant class and numbers of variants in cases and controls for each model would be helpful.
4. Overall the top results confirm previous exome sequencing and add several new signals. But many significant genes are identified in different models. It would be important to give QQ plots and top significant gene results for each of the 12 models. Some genes will be significant in just one model and others in multiple models.
5. The reported OR for most significant genes are lower than expected from previous studies. Odds ratios for the natural anticoagulant genes have been reported >8. Factor V Leiden >3.0. The lower ORs reported here could be due to the inappropriate inclusion of benign variants, the imprecise definition of cases and controls or some other factors. The authors should address this in the discussion section.
6. How do the authors define deleterious missense variants? This should also be stated in the results and the methods section for clarity. Were any of these definitions validated in vitro? Or are they all in silico predictions?
7. ABO and ADAMTS13 loci contain variants that are in LD. Please confirm that the ADAMTS13 signal is independent of ABO. Did the common variant analysis include only coding variants?
8. Factor V Leiden is a well studied common European variant that is associated with an increased lifetime risk of VTE. Investigators have long wondered why some FV Leiden carriers get VTE while others do not. Here the authors have the opportunity to look specifically at interactions between rare variants and FV Leiden. Do specific top signals in the gene collapsing analysis have a different frequency in Leiden carriers or non Leiden carriers? This would be an important focus of the excellent rare/common variant interaction already included in the manuscript.
9. Although large genomic studies routinely report predicted gene function and gene pathway studies, the results here are not novel or unexpected. Therefore I would consider using the supplement to include these studies in Figure 5 and 6.

10. The discussion section in its current form is not useful and obscures the important findings in this manuscript. I would avoid overinterpretation of the functional annotation studies as they are well known to be generalized as are the well documented pleiotropic effects of VTE variants. Discussion on the SRSF6, PHPT1 and MAP3K2 are very speculative and should be eliminated or limited to what is actually known about these gene functions.

Reviewer #2 (Remarks to the Author):

In the current study, He et al leverage WES data from the UK Biobank to perform the largest rare variant analysis of VTE to date. The authors leverage the available UK Biobank data to perform a variety of rare variant burden/collapsing tests to identify deleterious variants in genes that effect VTE risk. Overall, Sixteen known and four novel genes were identified through protein-coding variants, and replication in the FinnGen cohort is reported, though I have questions about this (see below). The authors nicely use this data in a PRS analysis to demonstrate the additive effects of rare and common variants on VTE risk, and perform a series of downstream analyses including PheWAS, pathway analyses via FUMA, a common variant GWAS in PLINK2, etc. In sum, some interesting observations are reported, however, this reviewer has several major and minor concerns.

Major concerns:

- 1) This reviewer's largest concern has to do with the reported replication in FinnGen. The authors provide no methodology for how this replication was performed, which is particularly relevant given that this analysis was used to replicate gene burden/rare variant analyses. Was the strongest associated protein coding variant used for FinnGen replication, or were gene-based tests performed? More detail is needed to explain how this replication was performed. It seems additionally important given that, as far as this reviewer can tell, the authors would theoretically be performing gene-based testing on imputed variants without access to individual level data. Furthermore, no direction of effect is provided for FinnGen.
- 2) In the annotation performed to allow for gene-based testing, multiple algorithms (SIFT, LRT, PolyPhen2 HDIV, and PolyPhen2 HVAR, and MutationTaster) are used. The the authors state that variants had to be "consistently" annotated by these algorithms, but no details are provided. Did it need to be called deleterious by 5/5 (ie, all) algorithms? This needs clarification
- 3) The authors report penetrance figures, but penetrance analyses are generally reserved for situations where we have granular clinical data to truly assess whether patient who carries a specific mutation has phenotypic evidence of disease. From what I can gather, the penetrance calculations here are all based on EHR data, which can be biased by misclassification associated with using procedure codes/ICD codes. This is particularly relevant given the small sample sizes associated with carriers of very rare variants. This reviewer suggests that the authors change the wording in this analysis.
- 4) The reported Bonferroni P value appears to only correct for the total number of genes tested. However, multiple different models are tested (at least 12), but these series of tests are not, from what this reviewer can gather, included in the Bonferroni correction. Can the authors state why this choice was made?
- 5) A gene "time to event" analysis is performed, but no specific details in the methods are clear about how this was performed. Were deleterious variants collapsed (this is what is implied in the discussion) - or some other analytic technique used? Further clarification is needed

Minor Concerns:

- 1) The common variant analysis offers very little in terms of novelty. Not sure it needs to be in the main manuscript (as opposed to the supplement)
- 2) Line 433, should this be minimum p value (instead of maximum)?

Reviewer #3 (Remarks to the Author):

The manuscript "Genetic associations of protein-coding variants in venous thromboembolism" details a whole exome sequencing genetic discovery study for identification of rare coding variants/genes associated with VTE using UK Biobank (self-reported Caucasian only) for discovery and the FinnGen consortium study as validation.

They find: (1) using gene burden analysis, several known VTE genes as well as a several novel VTE genes not previously implicated in thrombosis/coagulation; (2) using single common coding variant analysis, they find several novel genes for VTE; (3) rare variant carriage is additive to a common variant PRS; and (4) PheWAS showed mostly expected associations with clinical trials of thrombosis, hematologic abnormalities, etc.

The manuscript addresses an important question, namely novel genes associated with VTE given that the heritability of VTE remains unexplained. The manuscript is well written and the analyses appear to be well done and it is the largest WES study of VTE risk to date (however, a larger GWAS study has been done). The use of summary statistics for validation is also a strength.

Enthusiasm for the manuscript is significantly tempered due to the following: (1) use of a superficial VTE phenotype available in UKB (without details about how it was defined including separation of provoked vs. unprovoked VTE); (2) impact tempered given low effect sizes, not much novel information, and lack of functional insights; (3) lack of assessment of clinical confounders; and (4) lack of generalizability based on only utilizing individuals of European ancestry.

Additional concerns/questions include:

- (1) far too liberal of a p-value threshold of 0.01 for validation in FinnGen (major)
- (2) strong language around clinical implications of findings in discussion needs to be tempered
- (3) in the absence of including coding and non-coding variants, the analysis of common coding variants is not useful
- (4) a meta-analysis of UKB with FinnGen would be useful for understanding robustness of these findings in the context of other variants found in FinnGen
- (5) it is not clear what the purpose of the time-to-event analyses are in the context of the VTE case/control analyses (which are presumably inclusive of prevalent and incident VTE)
- (6) should not term it "penetrance" given the superficial EHR phenotyping used for VTE, instead can just term it "disease prevalence"
- (7) the additive results for rare variants on top of PRS should not be termed an "interaction" and in fact, the interaction term was not-significant. This is just additive risk.

The authors sincerely appreciate the critical reviews of the paper, and for the helpful way in which the reviewing editors put together a constructive list of suggestions for the revision of the paper. We have now revised the paper to carefully address all the points raised. Our responses below are preceded by “ - ”, and changes made to the paper are shown below within “...”, and in red font in the revised paper.

REVIEWER COMMENTS

Reviewer #1 (Remarks to the Author):

This is a nice manuscript that uses the resources of the UK Biobank to conduct a rare genetic variant analyses of venous thromboembolic disease. This allows the comparison of a large group of cases ~14,000 to an even larger group of controls >300,000. The authors employ up to date analysis pipelines and perform several ancillary studies to complement the results of their gene collapsing analyses. In addition to the collapsing analysis, I particularly enjoyed the addition of rare variants to the common variant polygenic risk score which demonstrated additive interaction with common variants. This study can be improved through a couple additional analyses, clarification and re-writing of the majority of the discussion section as outlined below.

Major points

1. In order to better understand the VTE cases and controls the authors must better explain the criteria they used to define VTE cases. I don't think this is anywhere in the manuscript. Were ICD9/10 codes used? How do the authors account for trauma, pregnancy, central line or cancer related VTE?

Response:

- Thank you for your rigorous consideration. We have added the detailed VTE definition in “**Method-Study population**” and **Supplementary Table 1**. (“**The**

VTE diagnosis in this project were based on self-report at baseline or electronic health records (EHR). We used the International Classification of Diseases system codes 10th revision (ICD-10) codes (I80.1, I80.2, I82.2, I26.0, and I26.9) and Office of Population and Censuses and Survey 4th Revision Procedures Codes (OPCS-4, L79.1 and L90.2) to identify VTE cases, which presented as a primary or secondary diagnosis in the hospital inpatient records or an underlying cause of death in the death register (**Supplementary Table 1**). Furthermore, we excluded individuals with diagnosis of superficial or unclear site of thrombophlebitis (I80.0, I80.3, I80.8, I80.9), portal vein thrombosis (I81), BuddChiari syndrome (I82.0), and known coagulation defects (D68) to minimize heterogeneity and bias^{9,63.}”)

2. No analyses are provided of the X-chromosome. This is not acceptable. Many coagulation genes are encoded for on the X-chromosome. Please use the UKBB and FinnGenn X-chromosome sequencing data.

Response:

- According to your suggestion, rare variants collapsing analysis and single variant analysis for X-chromosome were further conducted. We have included related information in **Methods** (“**For chromosome X, pseudoautosomal (PAR1 and PAR2) regions were further excluded in both sexes and heterozygous genotypes in males were set to missing^{21.}**”). However, there’s no significant genes or variants in X-chromosome survived after Bonferroni correction.

3. Although the authors define the 12 different collapsing models used for their primary analysis, this should be briefly explained in the results section, a simple table with allele frequency, variant class and numbers of variants in cases and controls for each model

would be helpful.

Response:

- According to your suggestion, we have added the details (allele frequency, variant class and numbers of variants) of 12 different collapsing models in **Supplementary Table 3** and explained it in the **Results** briefly. (“**Within each gene, qualifying variants (QV) containing either loss-of-function (LOF, stop gained, start lost, splice acceptor, splice donor, stop lost or frameshift) only, likely deleterious missense (Dmis, predicted to be deleterious by SIFT²², LRT²³, PolyPhen2 HDIV, and PolyPhen2 HVAR²⁴, and MutationTaster²⁵ consistently) only, or both LOF and Dmis were aggregated into distinct variant sets. These variant sets were created using QVs at four MAF thresholds ($<10^{-5}$, $<10^{-4}$, $<10^{-3}$, and $<10^{-2}$, Supplementary Table 3), and their impact on VTE was subsequently assessed.**”)

4. Overall the top results confirm previous exome sequencing and add several new signals. But many significant genes are identified in different models. It would be important to give QQ plots and top significant gene results for each of the 12 models. Some genes will be significant in just one model and others in multiple models.

Response:

- According to your suggestion, we have added Manhattan plots and QQ-plots for each of the 12 models in **Supplementary Fig. 1** and top 10 significant gene results for each of the 12 models in **Supplementary Table 5**.

5. The reported OR for most significant genes are lower than expected from previous studies. Odds ratios for the natural anticoagulant genes have been reported >8 . Factor V Leiden >3.0 .

The lower ORs reported here could be due to the inappropriate inclusion of benign variants, the imprecise definition of cases and controls or some other factors. The authors should address this in the discussion section.

Response:

- We greatly appreciate your astute observations and valuable insights. To minimize the possibility for inappropriate inclusion of benign variants and imprecise case and control definitions, we have implemented rigorous variant-level quality control¹, and our definition of VTE cases^{2,3} aligns with recent high-quality studies. Furthermore, we have provided other potential factors in our **Discussion** section. (“Notably, effect sizes for identified genes from collapsing analysis are lower than expected from previous studies. Considering that HR estimates from subsequent survival analysis were larger and more consistent with previous reports, we hypothesize that the comparatively smaller effect sizes observed during the initial gene discovery phase were primarily due to usage of saddle point-approximation-corrected logistic mixed-model approach implemented in SAIGE-GENE+ software, which might yield slightly conservative effect estimates, particularly when assessing significance for binary traits with imbalanced case-control ratios⁴⁰.”)

Reference:

- 1 Jurgens, S. J. *et al.* Analysis of rare genetic variation underlying cardiometabolic diseases and traits among 200,000 individuals in the UK Biobank. *Nat Genet* **54**, 240-250, doi:10.1038/s41588-021-01011-w (2022).
- 2 Klarin, D., Emdin, C. A., Natarajan, P., Conrad, M. F. & Kathiresan, S. Genetic Analysis of Venous Thromboembolism in UK Biobank Identifies the ZFPM2 Locus and Implicates Obesity as a Causal Risk Factor. *Circ Cardiovasc Genet*

10, doi:10.1161/circgenetics.116.001643 (2017).

- 3 Klarin, D. *et al.* Genome-wide association analysis of venous thromboembolism identifies new risk loci and genetic overlap with arterial vascular disease. *Nat Genet* **51**, 1574-1579, doi:10.1038/s41588-019-0519-3 (2019).

6. How do the authors define deleterious missense variants? This should also be stated in the results and the methods section for clarity. Were any of these definitions validated in vitro? Or are they all in silico predictions?

Response:

- Thank you for pointing out our rough description. Missense deleteriousness was predicted using five different in silico prediction algorithms, namely SIFT, LRT, PolyPhen2 HDIV, PolyPhen2 HVAR, and MutationTaster, and missense variants that were predicted to be deleterious by all five algorithms were annotated as ‘likely deleterious’ variants^{4,5}. We have added detailed description in both “**Results - Gene-level associations of rare coding variants with VTE**” (“**Within each gene, qualifying variants (QV) containing either loss-of-function (LOF, stop gained, start lost, splice acceptor, splice donor, stop lost or frameshift) only, likely deleterious missense (Dmis, predicted to be deleterious by SIFT²², LRT²³, PolyPhen2 HDIV, PolyPhen2 HVAR²⁴, and MutationTaster²⁵ consistently) only, or both LOF and Dmis were aggregated into distinct variant sets.**”) and “**Methods - Genotyping, QC, and variant annotation**” section. (“**Dmis variants (predicted to be deleterious by all five in silico algorithms, namely SIFT²², LRT²³, PolyPhen2 HDIV, PolyPhen2 HVAR²⁴, and MutationTaster²⁵)**”)

Reference:

- 4 Akbari, P. *et al.* Sequencing of 640,000 exomes identifies GPR75 variants

associated with protection from obesity. *Science* **373**, doi:10.1126/science.abf8683 (2021).

- 5 Rajagopal, V. M. *et al.* Rare coding variants in *CHRNA2* reduce the likelihood of smoking. *Nature Genetics* **55**, 1138-1148, doi:10.1038/s41588-023-01417-8 (2023).

7. *ABO* and *ADAMTS13* loci contain variants that are in LD. Please confirm that the *ADAMTS13* signal is independent of *ABO*. Did the common variant analysis include only coding variants?

Response:

- According to your suggestion, we checked carefully and confirmed that rs685523 (mapped to *ADAMTS13*) was in LD ($r^2 = 0.034$) with a more significantly associated SNP (rs8176719, mapped to *ABO*). We have redone the clumping procedure under a stricter clump window ($r^2 = 0.01$ and kb = 5000)⁶ to avoid the LD and changed corresponding “**Methods - Single variant analysis for common variants**” (“significant SNPs were selected to identify the lead SNPs within each risk locus, which were denoted as the SNPs with the lowest p-value when multiple SNPs were observed to be in strong LD ($r^2 > 0.01$) within a 5000-kb window.”).
- Our single variant analysis included only exonic coding variants of sufficient frequency (MAF > 0.01)^{7,8}. And we have added a sensitivity analysis including both coding and noncoding variants (**Supplementary Fig. 12**).

Reference:

- 6 Ghose, J. *et al.* Genome-wide meta-analysis identifies 93 risk loci and enables risk prediction equivalent to monogenic forms of venous thromboembolism. *Nat Genet* **55**, 399-409, doi:10.1038/s41588-022-01286-7 (2023).

- 7 Park, J. *et al.* Exome-wide association analysis of CT imaging-derived hepatic fat in a medical biobank. *Cell Rep Med* **3**, 100855, doi:10.1016/j.xcrm.2022.100855 (2022).
- 8 Choi, S. H. *et al.* Monogenic and Polygenic Contributions to Atrial Fibrillation Risk: Results From a National Biobank. *Circ Res* **126**, 200-209, doi:10.1161/circresaha.119.315686 (2020).

8. Factor V Leiden is a well studied common European variant that is associated with an increased lifetime risk of VTE. Investigators have long wondered why some FV Leiden carriers get VTE while others do not. Here the authors have the opportunity to look specifically at interactions between rare variants and FV Leiden. Do specific top signals in the gene collapsing analysis have a different frequency in Leiden carriers or non Leiden carriers? This would be an important focus of the excellent rare/common variant interaction already included in the manuscript.

Response:

- According to your suggestion, we have added the interaction analysis between rare coding variants and factor V Leiden variant rs6025 in “**Results - Interplay between genome-wide polygenic risk score (PRS) and rare coding variants**” section (“**We specifically explored the interaction between rare coding variants and factor V Leiden (F5) variant rs6025 (p.R534Q), a well-studied high-risk common variant, for which only *STAB2* showed additive effects (RERI=1.17, AP=0.30, S=1.67, **Supplementary Table 15**).**”).

9. Although large genomic studies routinely report predicted gene function and gene pathway studies, the results here are not novel or unexpected. Therefore I would

consider using the supplement to include these studies in Figure 5 and 6.

Response:

- According to your suggestion, we have put the content of Figure 5 and 6 into **Supplementary Tables 17-20** and Supplementary materials.

10. The discussion section in its current form is not useful and obscures the important findings in this manuscript. I would avoid overinterpretation of the functional annotation studies as they are well known to be generalized as are the well documented pleiotropic effects of VTE variants. Discussion on the *SRSF6*, *PHPT1* and *MAP3K2* are very speculative and should be eliminated or limited to what is actually known about these gene functions.

Response:

- According to your suggestion, we have deleted relevant content of the functional annotation and speculative description on *SRSF6*, *PHPT1* and *MAP3K2* in the discussion section. Please refer to the third and fourth paragraphs in the discussion.

Reviewer #2 (Remarks to the Author):

In the current study, He et al leverage WES data from the UK Biobank to perform the largest rare variant analysis of VTE to date. The authors leverage the available UK Biobank data to perform a variety of rare variant burden/collapsing tests to identify deleterious variants in genes that effect VTE risk. Overall, Sixteen known and four novel genes were identified through protein-coding variants, and replication in the FinnGen cohort is reported, though I have questions about this (see below). The authors nicely use this data in a PRS analysis to demonstrate the additive effects of rare and common variants on VTE risk, and perform a series of downstream analyses including

PheWAS, pathway analyses via FUMA, a common variant GWAS in PLINK2, etc. In sum, some interesting observations are reported, however, this reviewer has several major and minor concerns.

Major concerns:

1) This reviewer's largest concern has to do with the reported replication in FinnGen. The authors provide no methodology for how this replication was performed, which is particularly relevant given that this analysis was used to replicate gene burden/rare variant analyses. Was the strongest associated protein coding variant used for FinnGen replication, or were gene-based tests performed? More detail is needed to explain how this replication was performed. It seems additionally important given that, as far as this reviewer can tell, the authors would theoretically be performing gene-based testing on imputed variants without access to individual level data. Furthermore, no direction of effect is provided for FinnGen.

Response:

- We gratefully appreciate your critical comments. Indeed, we could only get access to GWAS summary results for VTE in FinnGen instead of individual level data. We have added more details in “**Methods-External replication in FinnGen**” section to explain how this replication was performed. Furthermore, we used Bonferroni correction to adjust for multiple comparisons problem for single variant analysis and gene-based analysis, respectively. (“**For gene-level collapsing analysis validation, we searched for variants with the strongest FinnGen VTE associations mapped to significant genes we found. For single variant analysis validation, we searched for our lead SNPs directly. Both approaches used Bonferroni correction with the number of significant genes or lead SNPs in that particular analysis.**”)
- According to your suggestion, we have provided the effect sizes of FinnGen

validation in revised **Table 1** and **Table 2**.

2) In the annotation performed to allow for gene-based testing, multiple algorithms (SIFT, LRT, PolyPhen2 HDIV, and PolyPhen2 HVAR, and MutationTaster) are used. The the authors state that variants had to be "consistently" annotated by these algorithms, but no details are provided. Did it need to be called deleterious by 5/5 (ie, all) algorithms? This needs clarification

Response:

- Missense deleteriousness was predicted using five different in silico prediction algorithms, namely SIFT, LRT, PolyPhen2 HDIV, PolyPhen2 HVAR, and MutationTaster, and missense variants that were predicted to be deleterious by all five algorithms were annotated as ‘likely deleterious’ variants^{4,5}. We have added detailed description in both “**Results**” (“**Within each gene, qualifying variants (QV) containing either loss-of-function (LOF, stop gained, start lost, splice acceptor, splice donor, stop lost or frameshift) only, likely deleterious missense (Dmis, predicted to be deleterious by SIFT²², LRT²³, PolyPhen2 HDIV, PolyPhen2 HVAR²⁴, and MutationTaster²⁵ consistently) only, or both LOF and Dmis were aggregated into distinct variant sets.**”) and “**Methods-Genotyping, QC, and variant annotation**” section (“**Dmis variants (predicted to be deleterious by all five in silico algorithms, namely SIFT²², LRT²³, PolyPhen2 HDIV, PolyPhen2 HVAR²⁴, and MutationTaster²⁵)**”).

Reference:

- 4 Akbari, P. *et al.* Sequencing of 640,000 exomes identifies GPR75 variants associated with protection from obesity. *Science* **373**, doi:10.1126/science.abf8683 (2021).

- 5 Rajagopal, V. M. *et al.* Rare coding variants in CHRN2 reduce the likelihood of smoking. *Nature Genetics* **55**, 1138-1148, doi:10.1038/s41588-023-01417-8 (2023).

3) The authors report penetrance figures, but penetrance analyses are generally reserved for situations where we have granular clinical data to truly assess whether patient who carries a specific mutation has phenotypic evidence of disease. From what I can gather, the penetrance calculations here are all based on EHR data, which can be biased by misclassification associated with using procedure codes/ICD codes. This is particularly relevant given the small sample sizes associated with carriers of very rare variants. This reviewer suggests that the authors change the wording in this analysis.

Response:

- According to your suggestion, we have reworded “penetrance” as “prevalence” in the manuscript.

4) The reported Bonferroni P value appears to only correct for the total number of genes tested. However, multiple different models are tested (at least 12), but these series of tests are not, from what this reviewer can gather, included in the Bonferroni correction.

Can the authors state why this choice was made?

Response:

- Since inherently low power is necessarily exacerbated when type 1 error is adjusted to control for multiple testing of all 12 collapsing models, we only used a Bonferroni correction for the total number of genes and made the significance threshold comparable with recent studies of exome sequencing⁸⁻¹⁰.

Reference:

- 8 Choi, S. H. *et al.* Monogenic and Polygenic Contributions to Atrial Fibrillation Risk: Results From a National Biobank. *Circ Res* **126**, 200-209, doi:10.1161/circresaha.119.315686 (2020).
- 9 Huang, Y. *et al.* Rare genetic variants impact muscle strength. *Nature Communications* **14**, 3449, doi:10.1038/s41467-023-39247-1 (2023).
- 10 Gao, X. R., Chiariglione, M. & Arch, A. J. Whole-exome sequencing study identifies rare variants and genes associated with intraocular pressure and glaucoma. *Nature Communications* **13**, 7376, doi:10.1038/s41467-022-35188-3 (2022).

5) A gene "time to event" analysis is performed, but no specific details in the methods are clear about how this was performed. Were deleterious variants collapsed (this is what is implied in the discussion) - or some other analytic technique used? Further clarification is needed.

Response:

- According to your suggestion, we have added the methodological detail of "time-to-event" analysis in "**Method - Time-to-event validation**". ("Within each significant gene in collapsing analysis, QVs contained in the model of strongest association with VTE were aggregated into a single variable to represent that gene. We also collapsed all significant genes, distinguishing carriers from noncarriers, to study combined effects of rare coding variants. For significant common variants, in addition to variant-level analysis, we calculated a PRS constructed using the LD-clumped lead SNPs by PLINK2 and divided the participants into 2 groups: low (standardized PRS<0) and high (standardized PRS>0) risk to represent combined

effects of common coding variants. Then Kaplan-Meier analyses and the CPH regression were conducted to investigate whether the survival probability of incident VTE differs substantially.”)

Minor Concerns:

1) The common variant analysis offers very little in terms of novelty. Not sure it needs to be in the main manuscript (as opposed to the supplement)

Response:

- We acknowledge your concern regarding the novelty. Although only two novel genes were identified through single variant analysis, they were consistently significant in subsequent longitudinal analyses, with even one validated in external FinnGen cohort. Furthermore, with the availability of WES data, exploring the genetic associations across the allele frequency spectrum directly without relying on imputation is important for providing a comprehensive understanding of the genetic architecture. Therefore, we believe that these findings warrant inclusion in the main manuscript rather than being relegated to supplementary materials.

2) Line 433, should this be minimum p value (instead of maximum)?

Response:

- Since LOVO analyses were performed by iteratively excluding each variant at a time from the variant sets in a gene and rerunning the association test. With the increasing importance of a variants, the association test will be less significant after removing it and P value will become larger. In other words, the most important variant within each gene was defined as the variant that was removed to achieve a maximum p-value. And we have explained it in more detail in the manuscript. “The

most important variant within each gene was defined as the variant that was removed to achieve a maximum association test p-value.”

Reviewer #3 (Remarks to the Author):

The manuscript “Genetic associations of protein-coding variants in venous thromboembolism” details a whole exome sequencing genetic discovery study for identification of rare coding variants/genes associated with VTE using UK Biobank (self-reported Caucasian only) for discovery and the FinnGen consortium study as validation.

They find: (1) using gene burden analysis, several known VTE genes as well as a several novel VTE genes not previously implicated in thrombosis/coagulation; (2) using single common coding variant analysis, they find several novel genes for VTE; (3) rare variant carriage is additive to a common variant PRS; and (4) PheWAS showed mostly expected associations with clinical trials of thrombosis, hematologic abnormalities, etc.

The manuscript addresses an important question, namely novel genes associated with VTE given that the heritability of VTE remains unexplained. The manuscript is well written and the analyses appear to be well done and it is the largest WES study of VTE risk to date (however, a larger GWAS study has been done). The use of summary statistics for validation is also a strength.

Enthusiasm for the manuscript is significantly tempered due to the following:

(1) use of a superficial VTE phenotype available in UKB (without details about how it was defined including separation of provoked vs. unprovoked VTE);

Response:

- Thank you for your rigorous consideration. We have added the detailed VTE definition in “**Method - Study population**” section (“**The VTE diagnosis in this project were based on self-report at baseline or electronic health records (EHR). We used the International Classification of Diseases system codes 10th revision (ICD-10) codes (I80.1, I80.2, I82.2, I26.0, and I26.9) and Office of Population and Censuses and Survey 4th Revision Procedures Codes (OPCS-4, L79.1 and L90.2) to identify VTE cases, which presented as a primary or secondary diagnosis in the hospital inpatient records or an underlying cause of death in the death register (Supplementary Table 1). Furthermore, we excluded individuals with diagnosis of superficial or unclear site of thrombophlebitis (I80.0, I80.3, I80.8, I80.9), portal vein thrombosis (I81), BuddChiari syndrome (I82.0), and known coagulation defects (D68) to minimize heterogeneity and bias^{8,9}.**”) and **Supplementary Table 1**. However, we were indeed unable to distinguish provoked from unprovoked VTE cases due to the insufficient data in UKB. We have added this point in the **limitation** section (“**Third, we were unable to identify whether VTE cases were provoked or unprovoked due to insufficient information in UKB.**”).

(2) impact tempered given low effect sizes, not much novel information, and lack of functional insights; **Response:**

- While we acknowledge the issue of relatively modest effect sizes in our results, we would like to emphasize several strengths of our research. First, we not only confirmed fourteen previously reported genes but also identified four novel genes associated with VTE, contributing to the novelty of our findings. In addressing the concern of small effect sizes, our **Discussion** section provides an in-depth analysis of this aspect (“**Notably, effect sizes for identified genes from collapsing analysis**

are lower than expected from previous studies. Considering that HR estimates from subsequent survival analysis were larger and more consistent with previous reports, we hypothesize that the comparatively smaller effect sizes observed during the initial gene discovery phase were primarily due to usage of saddle point-approximation-corrected logistic mixed-model approach implemented in SAIGE-GENE+ software, which might yield slightly conservative effect estimates, particularly when assessing significance for binary traits with imbalanced case-control ratios⁴⁰.”). Additionally, we have assessed potential biological functions of VTE-related genes by pathway enrichment analysis, which validated established pathways as previously reported and provided valuable insights into the mechanisms underpinning VTE.

(3) lack of assessment of clinical confounders;

Response:

- We deeply appreciate your attention to potential clinical confounders in our study. To minimize bias of clinical confounders, we have excluded individuals with diagnosis of superficial or unclear site of thrombophlebitis, portal vein thrombosis, BuddChiari syndrome, and known coagulation defects. The detailed VTE definition was added in “**Method - Study population**” section. Please see more details in response to reviewer 3’s comments (Major No.1). Furthermore, we adjusted for age, sex, and the top genetic principal components, which were commonly employed as covariates in recent genetic studies on VTE^{2,3,6,11-13}, to control potential clinical confounders.

Reference:

- 2 Klarin, D., Emdin, C. A., Natarajan, P., Conrad, M. F. & Kathiresan, S. Genetic

- Analysis of Venous Thromboembolism in UK Biobank Identifies the ZFPM2 Locus and Implicates Obesity as a Causal Risk Factor. *Circ Cardiovasc Genet* **10**, doi:10.1161/circgenetics.116.001643 (2017).
- 3 Klarin, D. *et al.* Genome-wide association analysis of venous thromboembolism identifies new risk loci and genetic overlap with arterial vascular disease. *Nat Genet* **51**, 1574-1579, doi:10.1038/s41588-019-0519-3 (2019).
- 6 Ghouse, J. *et al.* Genome-wide meta-analysis identifies 93 risk loci and enables risk prediction equivalent to monogenic forms of venous thromboembolism. *Nat Genet* **55**, 399-409, doi:10.1038/s41588-022-01286-7 (2023).
- 11 Desch, K. C. *et al.* Whole-exome sequencing identifies rare variants in STAB2 associated with venous thromboembolic disease. *Blood* **136**, 533-541, doi:10.1182/blood.2019004161 (2020).
- 12 Lindström, S. *et al.* Genomic and transcriptomic association studies identify 16 novel susceptibility loci for venous thromboembolism. *Blood* **134**, 1645-1657, doi:10.1182/blood.2019000435 (2019).
- 13 Thibord, F. *et al.* Cross-Ancestry Investigation of Venous Thromboembolism Genomic Predictors. *Circulation* **146**, 1225-1242, doi:10.1161/circulationaha.122.059675 (2022).

and (4) lack of generalizability based on only utilizing individuals of European ancestry.

Response:

- According to your suggestion, we have performed ancestry-specific and cross-ancestry meta-analysis of gene-level collapsing analysis and single variant analysis in non-British White (25,671 samples, 936 VTE cases), Asian (8558 samples, 194 VTE cases), Black (6628 samples, 251 VTE cases) and Mixed (5961 samples, 188

VTE cases) population, respectively. And we have added details in “**Methods - Multi-ancestry analysis**” and “**Results - Gene-level associations of rare coding variants with VTE**” (“In ancestry-specific analysis, the associations with nominal significance ($P < 0.05$) showed the same effect directions as in White British sample (**Supplementary Fig. 4**). As firm conclusions could not be drawn for some ethnic groups due to small sample sizes and low allele frequencies, cross-ancestry meta-analyses were performed. We found that the results were highly similar to those of only White British individuals (**Supplementary Fig. 5**, suggesting that the identified associations were not influenced by population stratification.”) and “**Results - Single variant associations of common coding variants with VTE**” section (“Moreover, the identified associations were also significant in the cross-ancestry meta-analysis (**Supplementary Figs. 9-10**) and were not influenced by population stratification as indicated by the high similarity with those of White British individuals (**Supplementary Fig. 11**).”). We also added this point in the **limitation** (“Second, as non-White British populations were under-represented in UKB, future studies with even larger sample sizes should be warranted in other ancestries.”).

Additional concerns/questions include:

(1) far too liberal of a p-value threshold of 0.01 for validation in FinnGen (major)

Response:

- According to your suggestion, we used Bonferroni correction to adjust for multiple comparisons problem for single variant analysis and gene-based analysis, respectively. And we have added more details in “**Methods - External replication in FinnGen**” (“Both approaches used Bonferroni correction with the number of

significant genes or lead SNPs in that particular analysis.”); “**Results - Gene-level associations of rare coding variants with VTE**” (“We also sought to confirm the exome-wide significant associations in FinnGen (release 8)¹³, and 5 of 6 identified genes were replicated ($P_{\text{Bonferroni}} < 0.05$) (**Table 1**).”) and “**Results - Single variant associations of common coding variants with VTE**” (“We also queried the GWAS summary results for VTE in FinnGen to validate the associations that we identified, and 12 of 13 significant associations were replicated ($P_{\text{Bonferroni}}$ ranged from 1.13×10^{-157} to 8.66×10^{-3} , **Table 2**).”)

(2) strong language around clinical implications of findings in discussion needs to be tempered

Response:

- According to your suggestion, we have tempered the strong language of the relevant clinical implications in **Discussion**: “~~Hence, our study indicates This supports~~ a possible relationship between *SRSF6* ~~gene-coagulation pathways induced by splicing machinery dysregulation~~ and VTE formation”; “our research work ... constitutes a valuable resource for thrombosis researchers and the discovery of new **potential** VTE therapeutic targets”; “~~These findings may be important This has great clinical implications, for example,~~ to inform future decisions on VTE screening”; “Some of the new genes may **be linked to contribute to VTE via well-characterized** coagulation pathways or ~~by influencing~~ hematological traits”.

(3) in the absence of including coding and non-coding variants, the analysis of common coding variants is not useful

Response:

- According to your suggestion, we have added sensitivity analysis of single variant analysis including both coding and non-coding variants in **Results** (“**Furthermore, ten out of 12 genes remained significant after inclusion of non-coding variants in the single variant analysis and linkage disequilibrium (LD) based clumping process (Supplementary Fig. 12).**”). However, the significant associations did not change substantially. Given that our study primarily focused on exonic coding variants, not much attention was paid to the investigation on noncoding variants. Notably, recent WES studies have also performed single variant analysis on common coding variants only^{7,8}, and we followed this practice.

(4) a meta-analysis of UKB with FinnGen would be useful for understanding robustness of these findings in the context of other variants found in FinnGen

Response:

- Thank you for your nice advice. However, we were unable to perform meta-analysis of UKB with FinnGen since we could only get access to GWAS summary results for VTE in FinnGen instead of individual level data. Our findings from WES data are not suitable for meta-analysis, especially for the gene-level associations. We acknowledged that FinnGen is not an ideal cohort to perfectly replicate the results from exome data, but it still can provide evidence to support our results with reference to recent whole exome studies^{10,14}. We also added this point in the **limitation** (“**Furthermore, GWAS summary data from FinnGen is not ideal enough for perfectly replicating our results from exome sequencing data, However, it can still provide evidence to support our results.**”).

Reference:

- 10 Gao, X. R., Chiariglione, M. & Arch, A. J. Whole-exome sequencing study identifies rare variants and genes associated with intraocular pressure and glaucoma. *Nature Communications* **13**, 7376, doi:10.1038/s41467-022-35188-3 (2022).
- 14 Nag, A. *et al.* Human genetics uncovers MAP3K15 as an obesity-independent therapeutic target for diabetes. *Sci Adv* **8**, eadd5430, doi:10.1126/sciadv.add5430 (2022).

(5) it is not clear what the purpose of the time-to-event analyses are in the context of the VTE case/control analyses (which are presumably inclusive of prevalent and incident VTE)

Response:

- Our gene-level collapsing analysis and single variant analysis including both prevalent and incident VTE, which precluded causal interpretations with its cross-sectional design and cannot determine the temporality of associations. Such cross-sectional analysis needs longitudinal study with a prospective design (excluding prevalent VTE cases) to confirm whether the result is robust to reverse confounding. And we have added the methodological detail in “**Method - Time-to-event validation**” (“**Time zero was the date of recruitment to UKB**, and follow-up time was subsequently calculated as years from it to the date of first diagnosis, death, or the final date with accessible information from hospital admission, whichever came first. We **excluded 48,515 participants with a VTE diagnosis before recruitment or without follow-up**, and the remaining **300,523 participants (6,920 VTE cases and 293,603 controls)** were included in the longitudinal analysis.”).

(6) should not term it “penetrance” given the superficial EHR phenotyping used for VTE, instead can just term it “disease prevalence”

Response:

- According to your suggestion, we have reworded “penetrance” as “prevalence” in the manuscript given the possible misclassification in EHR-derived phenotypes.

(7) the additive results for rare variants on top of PRS should not be termed an “interaction” and in fact, the interaction term was not-significant. This is just additive risk.

Response:

- According to your suggestion, we have reworded “interaction” into “additive effects” in the **Results** (“we observed a significant **additive effects**”), and Methods (“Additive **effects** were measured as the RERI, S, and AP due to interaction using epiR package”).

References

- 1 Jurgens, S. J. *et al.* Analysis of rare genetic variation underlying cardiometabolic diseases and traits among 200,000 individuals in the UK Biobank. *Nat Genet* **54**, 240-250, doi:10.1038/s41588-021-01011-w (2022).
- 2 Klarin, D., Emdin, C. A., Natarajan, P., Conrad, M. F. & Kathiresan, S. Genetic Analysis of Venous Thromboembolism in UK Biobank Identifies the ZFPM2 Locus and Implicates Obesity as a Causal Risk Factor. *Circ Cardiovasc Genet* **10**, doi:10.1161/circgenetics.116.001643 (2017).
- 3 Klarin, D. *et al.* Genome-wide association analysis of venous thromboembolism identifies new risk loci and genetic overlap with arterial vascular disease. *Nat*

- Genet* **51**, 1574-1579, doi:10.1038/s41588-019-0519-3 (2019).
- 4 Akbari, P. *et al.* Sequencing of 640,000 exomes identifies GPR75 variants associated with protection from obesity. *Science* **373**, doi:10.1126/science.abf8683 (2021).
- 5 Rajagopal, V. M. *et al.* Rare coding variants in CHRNA2 reduce the likelihood of smoking. *Nature Genetics* **55**, 1138-1148, doi:10.1038/s41588-023-01417-8 (2023).
- 6 Ghose, J. *et al.* Genome-wide meta-analysis identifies 93 risk loci and enables risk prediction equivalent to monogenic forms of venous thromboembolism. *Nat Genet* **55**, 399-409, doi:10.1038/s41588-022-01286-7 (2023).
- 7 Park, J. *et al.* Exome-wide association analysis of CT imaging-derived hepatic fat in a medical biobank. *Cell Rep Med* **3**, 100855, doi:10.1016/j.xcrm.2022.100855 (2022).
- 8 Choi, S. H. *et al.* Monogenic and Polygenic Contributions to Atrial Fibrillation Risk: Results From a National Biobank. *Circ Res* **126**, 200-209, doi:10.1161/circresaha.119.315686 (2020).
- 9 Huang, Y. *et al.* Rare genetic variants impact muscle strength. *Nature Communications* **14**, 3449, doi:10.1038/s41467-023-39247-1 (2023).
- 10 Gao, X. R., Chiariglione, M. & Arch, A. J. Whole-exome sequencing study identifies rare variants and genes associated with intraocular pressure and glaucoma. *Nature Communications* **13**, 7376, doi:10.1038/s41467-022-35188-3 (2022).
- 11 Desch, K. C. *et al.* Whole-exome sequencing identifies rare variants in STAB2 associated with venous thromboembolic disease. *Blood* **136**, 533-541, doi:10.1182/blood.2019004161 (2020).

- 12 Lindström, S. *et al.* Genomic and transcriptomic association studies identify 16 novel susceptibility loci for venous thromboembolism. *Blood* **134**, 1645-1657, doi:10.1182/blood.2019000435 (2019).
- 13 Thibord, F. *et al.* Cross-Ancestry Investigation of Venous Thromboembolism Genomic Predictors. *Circulation* **146**, 1225-1242, doi:10.1161/circulationaha.122.059675 (2022).
- 14 Nag, A. *et al.* Human genetics uncovers MAP3K15 as an obesity-independent therapeutic target for diabetes. *Sci Adv* **8**, eadd5430, doi:10.1126/sciadv.add5430 (2022).

REVIEWER COMMENTS

Reviewer #1 (Remarks to the Author):

The authors provide a revision to their original submission. In this revision, they adequately address this reviewer's concerns and modify the manuscript appropriately. The manuscript now includes important details about phenotyping methods and adds important analyses of common and rare variants.

I have no further critiques.

Reviewer #2 (Remarks to the Author):

I have reviewed the comments and responses made by the authors, and most of them have adequately addressed my concerns. However, the use of FinnGen summary statistics/summary level data for replication of gene-based testing is not adequately explained, and I have concerns that the methods here are not appropriate. Gene-based tests (such as those "collapsing" deleterious variants) require individual level, exome sequencing (or, in the past exome array) data to perform appropriately. The methods used in the UKBB data appear appropriate, but the explanation for how this is performed in the FinnGen summary statistics is: "For gene-level collapsing analysis validation, we searched for variants with the strongest FinnGen VTE associations mapped to significant genes we found."

There are methods published that outline potential strategies for using summary statistics for gene-based testing, though many have substantial weaknesses. Which of these were used? This section needs further clarification. The text as written suggests that any variant near the gene of interest (coding or non coding) could be used for replication. Some sort of method is necessary here to refine the replication strategy to be sure that the association signal in a given gene identified in UKBB is being replicated in FinnGen in that same gene. If only a single, coding variant was used for replication, this should be mentioned and clearly stated that a gene-based test was not performed.

The authors sincerely appreciate the critical reviews of the paper, and for the helpful way in which the reviewing editors put together a constructive list of suggestions for the revision of the paper. We have now revised the paper to carefully address all the points raised. Our responses below are preceded by “ - ”, and changes made to the paper are shown below within “...”, and in red font in the revised paper.

Detailed Response to Reviewers

Reviewer #1: (Remarks to the Author):

The authors provide a revision to their original submission. In this revision, they adequately address this reviewers concerns and modify the manuscript appropriately. The manuscript now includes important details about phenotyping methods and adds important analyses of common and rare variants.

I have no further critiques.

Response:

- Thank you very much!

Reviewer #2 (Remarks to the Author):

I have reviewed the comments and responses made by the authors, and most of them have adequately addressed my concerns. However, the use of FinnGen summary statistics/summary level data for replication of gene-based testing is not adequately explained, and I have concerns that the methods here are not appropriate. Gene-based tests (such as those "collapsing" deleterious variants) require individual level, exome sequencing (or, in the past exome array) data to perform appropriately. The methods used in the UKBB data appear appropriate, but the explanation for how this is performed in the FinnGen summary statistics is: "For gene-level collapsing analysis

validation, we searched for variants with the strongest FinnGen VTE associations mapped to significant genes we found."

There are methods published that outline potential strategies for using summary statistics for gene-based testing, though many have substantial weaknesses. Which of these were used? This section needs further clarification. The text as written suggests that any variant near the gene of interest (coding or non coding) could be used for replication. Some sort of method is necessary here to refine the replication strategy to be sure that the association signal in a given gene identified in UKBB is being replicated in FinnGen in that same gene. If only a single, coding variant was used for replication, this should be mentioned and clearly stated that a gene-based test was not performed.

Response:

- Thank you for your valuable feedback provided regarding the use of FinnGen summary statistics for gene-based testing. In response, we have further performed a gene-based association analysis using the mBAT-combo method implemented in GCTA software, specifically tailored for GWAS summary statistics. This approach enhances our replication strategy by effectively integrating multiple association signals within genes. Detailed methodology have been added to the “**Methods-External replication in FinnGen**” section (“**We further performed a gene-based association analysis using the summary statistics as input in GCTA software, with “mBAT-combo” command combining multi-SNP statistics effectively through a Cauchy combination method⁷⁸.**”) and “**Results-Gene-level associations of rare coding variants with VTE**” (“**We also sought to confirm the exome-wide significant associations in FinnGen (release 8)²⁶. Of the six identified genes, five replicated when searching for the most significant variants mapped to each gene, while four were validated by our 'mBAT-combo' gene-based analysis**

($P_{\text{Bonferroni}} < 0.05$, **Table 1**).”).

- We acknowledged that FinnGen is not an ideal cohort to perfectly replicate the results from exome data, but it still can provide evidence to support our results with reference to recent whole exome studies^{1,2}. We also added this point in the **limitation** (“Furthermore, GWAS summary data from FinnGen is not ideal enough for perfectly replicating our results from exome sequencing data, However, it can still provide evidence to support our results.”).

1 Nag, A. *et al.* Human genetics uncovers MAP3K15 as an obesity-independent therapeutic target for diabetes. *Sci Adv* **8**, eadd5430, doi:10.1126/sciadv.add5430 (2022).

2 Gao, X. R., Chiariglione, M. & Arch, A. J. Whole-exome sequencing study identifies rare variants and genes associated with intraocular pressure and glaucoma. *Nature Communications* **13**, 7376, doi:10.1038/s41467-022-35188-3 (2022).

REVIEWERS' COMMENTS

Reviewer #2 (Remarks to the Author):

I have no further comments

The authors are pleased to hear the positive responses from the reviewers, and sincerely appreciate their time and effort.